EMBO
reports

# Evolutionary relaxation and functional change of INSL3 and RXFP2 may underlie natural cryptorchidism in mammals

Yu Zheng [1], Simin Chai[2], Cuijuan Zhong [1], Yixuan Sun[1], Shixia Xu[1], Wenhua Ren[1] & Guang Yang [1,2]✉

## Abstract

**Cryptorchidism is a common congenital abnormality that increases infertility and testicular cancer risk in adulthood. However, a few mammals exhibit naturally undescended testes while maintaining normal reproduction. The mechanisms underlying this natural cryptorchidism remain unclear. Here, we found evolutionary relaxation in *INSL3* and *RXFP2* of cryptorchid mammals, with the highest dN/dS ratio observed in cetaceans. Cellular experiments demonstrated that cetacean INSL3 downregulated the cAMP-PKA-CREB pathway, thereby reducing gubernacular cell proliferation and contraction. Cetacean INSL3 knock-in mice exhibited groin-located testes, nearly perfectly mimicking cryptorchid phenotypes in cetaceans and other mammals with incompletely descended testes. Collagen and muscle fibers in the gubernaculum of transgenic mice were reduced, with differentially expressed genes enriched in muscle development and contraction pathways. Additionally, the knock-in mice displayed male sterility, impaired testicular development, and upregulated inflammatory pathways in the testes. Our findings reveal how evolutionary changes in the INSL3/RXFP2 pathway contribute to natural cryptorchidism in mammals and provide insights for investigating reproductive health and cancer resistance in cryptorchid species.**

**Keywords** Cryptorchidism; INSL3; Gubernaculum Development; Inflammation in Cryptorchid Testes; Evolutionary Relaxation
**Subject Categories** Development; Evolution & Ecology; Genetics, Gene Therapy & Genetic Disease

## Introduction

The testis is central to the male reproductive system, and thus, descent of the testes into the temperate environment of the scrotum represents a crucial developmental step toward successful reproduction for most mammals (Kleisner et al, 2010). Generally, the testes originate in the abdomen in all male creatures and gradually descend during development (Miller and Torday 2019). In most mammals, failure of testicular descent into the scrotum constitutes a pathological condition that impairs male fertility. This condition, known as cryptorchidism or undescended testis, was defined by the permanent failure of one or both testes to descend and is one of the most common congenital abnormalities in boys (Group JRHCS 1992; Li et al, 2024). Notably, cryptorchidism was recognized as one of the few established risk factors for both testicular cancer and subfertility (Gurney et al, 2017). In contrast, in birds and certain mammals, the testes remained near the kidneys, while in other mammals, they descended to varying extents until reaching their final mature position. For example, monotremes and most afrotherians had high intra-abdominal undescended testes in the same initial position as the ovaries in females. In addition, xenarthrans and several lineages of Boreoeutheria (e.g., cetaceans, flying foxes, some pinnipeds, eulipotyphlans, and certain rodents) possessed incompletely descended testes and lacked a scrotum (Kleisner et al, 2010; Werdelin and Nilsonne, 1999). In other words, some mammals exhibited naturally undescended testes, yet retained normal reproductive function in a cryptorchid state. However, the mechanism behind this natural "cryptorchidism" state has not been well revealed so far.

Although it is not feasible to experimentally verify the mechanism of testicular descent across all mammalian species, it is generally assumed that the underlying pathways are conserved, based on extensive studies in model organisms such as humans, mice, and a few other eutherians. The process of testicular descent, extensively studied in humans and mice, was widely recognized to occur in two hormonally distinct phases (Hutson et al, 2015). The first, transabdominal phase, involved the testes moving from their origin near the kidneys to the entrance of the inguinal canal, primarily driven by Insulin-like factor 3 (INSL3) signaling on the gubernaculum, which persists even in naturally cryptorchid mammals. The second, inguinoscrotal phase, involved the testes descend through the inguinal canal into the scrotum under the influence of androgens (Laptoiu et al, 2023; Sarila et al, 2022). INSL3, produced by Leydig cells, was first identified during studies on testicular descent (Adham et al, 1993). Through binding to relaxin family peptide receptor 2 (RXFP2), INSL3 induced collagen

[1]Jiangsu Key Laboratory for the Biodiversity Conservation and Sustainable Utilization in the Middle and Lower Reaches of the Yangtze River Basin, College of Life Sciences, Nanjing Normal University, Nanjing 210023, China. [2]Southern Marine Science and Engineering Guangdong Laboratory (Guangzhou), Guangzhou, Guangdong 511458, China.
✉E-mail: gyang@njnu.edu.cn

accumulation and swelling in the distal gubernaculum, allowing space in the abdominal cavity for testis descent (Emmen et al, 2000; Hrabovszky et al, 2002; Hutson et al, 2009; Lie and Hutson, 2011). Concurrently, testosterone induced regression of the cranial suspensory ligament, reducing resistance to descent. Together, these processes moved the testes toward the inguinal canal, where CGRP-stimulated contractions guided them into the scrotum to complete the descent (Yong et al, 2008). In mice, *Insl3* gene deletion caused high intra-abdominal cryptorchidism, demonstrating the essential role of INSL3 in gubernacular development and testicular descent (Bogatcheva et al, 2003; Parada, 1999). Studies in mice lacking RXFP2 had confirmed this receptor as INSL3's sole mediator in vivo, with similar gene mutations identified in cryptorchid patients (Canto et al, 2003; Dicke et al, 2023; El Houate et al, 2007). Beyond these findings in model organisms, recent comparative genomic studies in afrotherian mammals further underscore the evolutionary significance of this pathway: multiple independent pseudogenization events in *INSL3* and *RXFP2* were found to coincide with testicond phenotypes, suggesting strong evolutionary associations with cryptorchidism (Sharma et al, 2018). However, the broader role of INSL3 in evolutionary developmental mechanisms of testicular positioning across mammals remains to be clarified, highlighting a notable area for further research.

To test the hypothesis that INSL3 maintains natural cryptorchidism in mammals, we conducted evolutionary analyses, in combination with in vitro and in vivo experiments on INSL3 and its receptor RXFP2 of cetaceans, a typically representative group of natural cryptorchid. We found evidence of relaxed selection in *INSL3* within cryptorchid species, and cetacean *INSL3* introduced into mice induced inguinal cryptorchidism with gubernacular regression. Notably, transgenic model mice didn't maintain normal reproductive function post-cryptorchidism, with further testicular transcriptomic analysis showing downregulation of pathways essential for spermatogenesis. These findings could provide new insights into the mechanisms of natural cryptorchidism in mammals, and the constructed cryptorchid mouse model could serve as a valuable tool for future investigation on the reproductive health maintenance and tumor suppression under cryptorchid conditions.

# Results

## INSL3 and RXFP2 genes experience relaxed selection in cryptorchid mammals

Based on the evolutionary analyses of 46 representative mammals, it was shown that both *INSL3* and *RXFP2* exhibited dN/dS ratios below 1 across mammals, with higher ratios in cryptorchid mammals compared to scrotal mammals (Table EV1). In cetaceans and afrotherians with intact coding sequences for both genes, simultaneous relaxed selection of *INSL3* and *RXFP2* genes was observed when examined separately within the cryptorchidism lineage (Fig. 1A). Notably, cetaceans and the hippopotamus exhibited the highest *INSL3* dN/dS ratios among cryptorchid mammals, although *RXFP2* in the hippopotamus did not show a similar increase (Fig. 1B; Table EV1). RELAX model analysis also revealed significant relaxation of selective pressure on *INSL3* in cryptorchid mammals relative to scrotal mammals (K = 0.31,

$p = 0.0003$). Phylogenetic generalized least squares (PGLS) regression revealed a significant positive correlation between *INSL3* evolutionary rate and testicular descent ($p = 0.03341$, $r^2 = 0.08193$, $\lambda = 1$), despite no relevant convergent/parallel amino acid substitutions being identified between cryptorchid mammals.

## Cetacean INSL3 inadequately activates the cAMP-PKA-CREB signaling pathway, reducing proliferation and contraction in gubernacular cells

To investigate functional changes in cetacean INSL3, we first confirmed its interaction with mouse RXFP2 through co-immunoprecipitation (Fig. EV1). ELISA quantification of cAMP generated after treating with cetacean INSL3 and mouse INSL3 showed no significant differences, fluctuating within a similar range (Fig. 1C). Subsequent analysis of downstream PKA showed a time-dependent increase in levels with no significant differences between two groups; however, PKA activity was significantly reduced overall following cetacean INSL3 treatment, with a more rapid decline observed after 30 min (Fig. 1D). Western blotting (WB) analysis showed that CREB phosphorylation levels downstream were reduced in the cetacean INSL3-treated group compared to the control (Fig. 1E).

We treated primarily cultured mouse gubernacular cells with different concentrations of INSL3. CCK-8 analysis showed that cetacean INSL3 consistently had lower proliferative effects on these cells compared to mouse INSL3, with optimal effects observed at $10^{-8}$M for both. Subsequent growth curve analysis at $10^{-8}$M revealed that cetacean INSL3 continued to significantly slow gubernacular cell proliferation after 48 h (Fig. 2A). Using the fluorescent phalloidin marker for F-actin, we observed that gubernacular cells treated with mouse INSL3 exhibited increased F-actin content and more organized microfilament distribution, whereas the cetacean INSL3-treated group showed a smaller increase in F-actin levels (Fig. 2B).

## Cryptorchidism is present in cetacean INSL3 gene knock-in mice

Functional alterations in cetacean INSL3 provided a unique opportunity to examine the role of INSL3 in the diversity of testicular positioning among mammals. We generated cetacean *INSL3* knock-in (*INSL3*-KI) mice by replacing the full-length coding sequence of cetacean *INSL3* into the endogenous mouse locus, under the control of native regulatory elements. This strategy allowed us to assess the functional capacity of the cetacean ligand in the absence of endogenous mouse *INSL3*. To assess the testicular position in sexually mature *INSL3*-KI mice, with wildtype (WT) and *INSL3* knockout (*INSL3*-KO) mice as controls, it was observed that *INSL3*-KI mice exhibited a distinct testicular positioning, with the testes located in the inguinal region. This differs from WT mice, whose testes descend into the scrotum, and *INSL3*-KO mice, whose testes remain undescended near the kidney. Although cetacean testes are intra-abdominal and located dorsally and caudally, closely apposed to the lumbar body wall and posterior to the kidneys (Rommel et al, 1992), the intermediate positioning in *INSL3*-KI mice phenotypically reflects the partial descent characteristic of naturally cryptorchid mammals (Fig. 3A). We measured the anogenital distance (AGD) to quantify male characteristics in cryptorchid mice. Before sexual maturation, only

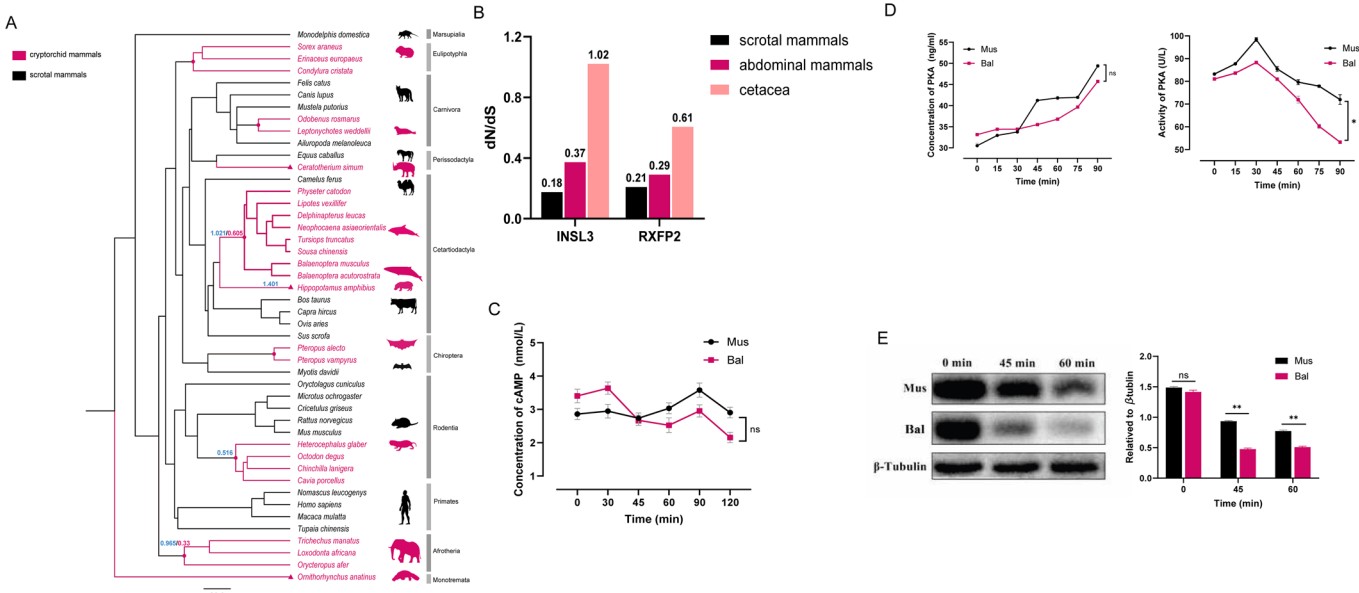

**Figure 1. dN/dS ratios of *INSL3* and *RXFP2* in different lineages and effects of cetacean INSL3 on downstream molecules.**

Figure (A) shows the phylogenetic relationships of the selected species, with numbers at nodes (*INSL3* in blue, *RXFP2* in red) representing the ω (dN/dS) values along cryptorchidism lineages where relaxed selection has occurred. (B) dN/dS ratios of *INSL3* and *RXFP2* in different taxa. Higher ratios indicate reduced purifying selection or potential shifts in selective pressure, rather than absolute evolutionary rates. (C) Time-dependent changes in cellular cAMP levels following different INSL3 treatments. (D) Time-dependent changes in cellular PKA content and activity following different INSL3 treatments. (E) Changes in CREB phosphorylation levels in cells treated with different INSL3. "Bal" indicates treatment with recombinant INSL3 protein from *Balaenoptera acutorostrata* (minke whale), and "Mus" indicates treatment with recombinant INSL3 protein from *Mus musculus* (mouse). (Data were shown as mean ± s.e.m. from $n = 3$ independent biological replicates, each with three technical replicates. Statistical significance was determined using two-tailed *t*-tests. Exact *p* values: (C) Mus vs Bal, $p = 0.4309$; (D) concentration Mus vs Bal, $p = 0.3054$; activity Mus vs Bal, $p = 0.0063$; (E) 0 min Mus vs Bal, $p = 0.0533$; 45 min Mus vs Bal, $p < 0.0001$; 60 min Mus vs Bal, $p < 0.0001$. *$p < 0.01$; **$p < 0.001$; ns not significant). Source data are available online for this figure.

*INSL3*-KO mice showed a significantly shortened AGD. However, after maturation, *INSL3*-KI mice, like *INSL3*-KO mice, exhibited a markedly reduced AGD compared to WT mice, suggesting a persistent impairment (Fig. 3B).

To investigate the cause of inguinal cryptorchidism in *INSL3*-KI mice, we observed the gubernaculum and stained tissue sections from postnatal day 15 (P15) mice. In contrast to WT mice, whose gubernaculum was enlarged and differentiated into muscle, the *INSL3*-KI mice showed a still-elongated gubernaculum with reduced muscle and collagen content (Fig. 3C).

To better model the natural cetacean condition, we generated cetacean *RXFP2* knock-in mice (*RXFP2*-KI) and bred them with *INSL3* knock-ins to produce double homozygous mice (*INSL3/RXFP2*-DKI). Upon examining *RXFP2*-KI and *INSL3/RXFP2*-DKI mice for testicular position and gubernacular development, we observed that both displayed testicular positioning in the groin, similar to *INSL3*-KI mice. Moreover, quantification showed that the testes in *INSL3/RXFP2*-DKI mice were slightly lower than in single knock-ins. Additionally, *RXFP2*-KI and *INSL3/RXFP2*-DKI mice exhibited undescended, shortened gubernaculum with reduced muscle and collagen content (Fig. EV2).

## Cetacea INSL3 induced changes in the gubernaculum transcriptome

RNA-seq analysis of the gubernaculum revealed distinct transcriptomic alterations in *INSL3*-KI and *INSL3*-KO mice relative to WT

controls (Fig. 4A). At postnatal day 8 (P8), both *INSL3*-KI and *INSL3*-KO mice exhibited downregulation of muscle development and cytoskeletal organization genes (Fig. 4B). However, *INSL3*-KO mice showed additional strong downregulation of Notch and AMPK signaling components (Fig. 4C), suggesting that cetacean INSL3 partially retains RXFP2 signaling capacity and mitigates some transcriptional defects seen in the *INSL3*-KO condition. This distinction is visually apparent in the P8 gubernaculum transcriptome overview (Fig. 4D), where the scatter plot and marginal densities reveal gene sets uniquely altered in *INSL3*-KO or *INSL3*-KI mice, and the Venn diagram shows only partial overlap of significantly differentially expressed genes between the two comparisons.

At postnatal day 23 (P23), the *INSL3*-KO gubernaculum was too severely degenerated and reduced in size to permit RNA isolation, whereas *INSL3*-KI mice displayed a marked increase in downregulated genes compared to P8 (Fig. 4A). KEGG enrichment analysis revealed that downregulated genes in the P23 *INSL3*-KI gubernaculum were predominantly associated with calcium, MAPK, cAMP, and PI3K–Akt signaling pathways. (Fig. 4E). Gene ontology (GO) enrichment analysis of these downregulated genes highlighted muscle structure and function-related processes, such as sarcomere organization, muscle contraction, and myofibril assembly, consistent with impaired gubernacular function during testicular descent (Fig. 4F). Given the known roles of calcium, MAPK, and cAMP pathways in testicular descent, we further examined DEGs within these cascades at both P8 and P23.

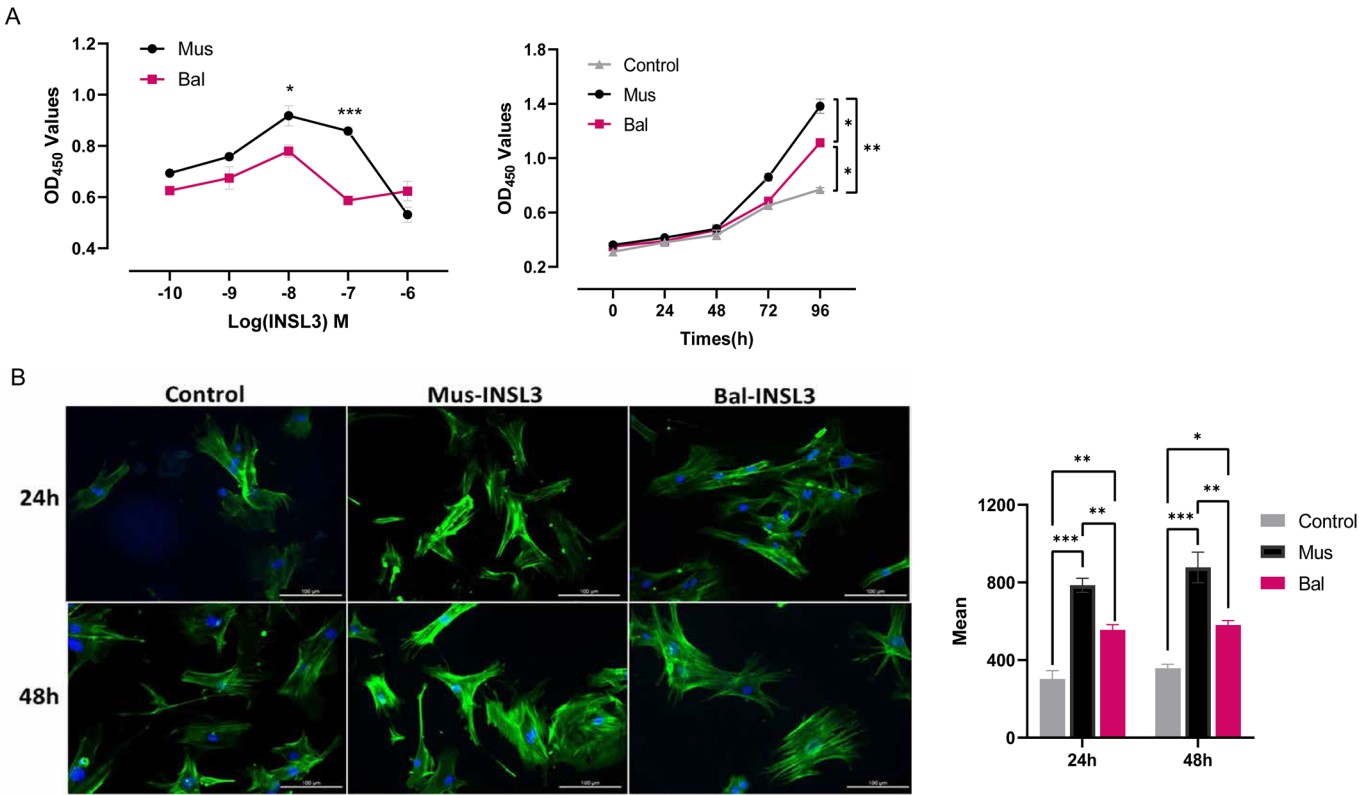

**Figure 2. Effect of cetacean INSL3 on gubernaculum cells.**

(A) Gubernacular cell activity at two INSL3 concentration gradients and cell proliferation at the optimal concentration. (Data were shown as mean ± s.e.m. from $n = 3$ biological replicates. Statistical significance was determined using two-tailed $t$-tests. Exact $p$ values: -8 Mus vs Bal, $p = 0.0401$; -7 Mus vs Bal, $p = 0.0003$; 96 h Control vs Mus, $p = 0.0017$; 96 h Control vs Bal, $p = 0.0102$; 96 h Mus vs Bal, $p = 0.0126$. *$p < 0.05$; **$p < 0.01$; ***$p < 0.001$). (B) Changes of F-actin in gubernacular cells after different INSL3 treatments. Scale bar, 100 µm. (Data were shown as mean ± s.e.m. from $n = 3$ biological replicates. Statistical significance was determined using ANOVA tests. Exact $p$ values: 24 h Control vs Mus, $p = 0.0001$; 24 h Control vs Bal, $p = 0.0021$; 24 h Mus vs Bal, $p = 0.0038$; 48 h Control vs Mus, $p = 0.0001$; 48 h Control vs Bal, $p = 0.0109$; 48 h Mus vs Bal, $p = 0.0041$. *$p < 0.05$; **$p < 0.01$; ***$p < 0.001$). "Bal" indicates treatment with recombinant INSL3 protein from *Balaenoptera acutorostrata* (minke whale), and "Mus" indicates treatment with recombinant INSL3 protein from *Mus musculus* (mouse). Source data are available online for this figure.

Heatmaps revealed pronounced changes in *INSL3*-KI mice, supporting the idea that disruption of these signaling networks contributes to the observed gubernacular abnormalities and cryptorchid phenotype (Fig. 4G).

## Cryptorchidism in *INSL3*-KI mice impairs their fertility potential

A 6-month fertility assessment of *INSL3*-KI mice showed that homozygous males were completely infertile, while the fertility of heterozygous males and females remained unaffected (Fig. 5A). Testes of *INSL3*-KI mouse were characterized at different ages. At postnatal day 20 (P20), *INSL3*-KI mice showed a slight, non-significant reduction in testicular weight, while *INSL3*-KO mice exhibited a significant reduction. At postnatal day 40 (P40) and postnatal day 60 (P60), *INSL3*-KI testes were up to 70% lighter than those of WT mice (Fig. 5B). The testicular structure of the seminiferous tubules (STs) was assessed and classified as normal (all germ cell layers present and correctly positioned), abnormal (germ cell loss or misplacement), or Sertoli cell-only (SCO, no germ cells). No variation in the percentage of normal seminiferous tubules was observed in WT mice across all ages. However, at P20, a few abnormal seminiferous tubules were

already observable in the testes of *INSL3*-KO mice. In addition, seminiferous tubule abnormalities in *INSL3*-KI mice did not appear until P40, at which point nearly all tubules in *INSL3*-KO mice were already abnormal. At P60, seminiferous tubules in both *INSL3*-KO and *INSL3*-KI mice predominantly degenerated into SCO (Fig. 5C,D). Additionally, since the tight and adherens junctions between Sertoli cells are crucial components of the blood-testis barrier, they facilitate the migration of developing germ cells from the basal to adluminal compartments of the seminiferous tubules. In *INSL3*-KI mice, however, Sertoli cell nuclei were frequently located towards the lumen of the seminiferous tubules with associated germ cell disorganization. Meanwhile, mature spermatozoa were reduced in the epididymis of *INSL3*-KI mice compared to WT mice. Analysis using the computer-assisted sperm analysis (CASA) system further revealed a severely reduced sperm motility in *INSL3*-KI mice (Fig. 5E,F).

## Transcriptomic alterations in the testes induced by cetacean INSL3

RNA-seq analysis was performed on testes from WT, *INSL3*-KI, and *INSL3*-KO mice at three key developmental stages (P8, P23, and P40) to investigate the molecular basis of infertility in

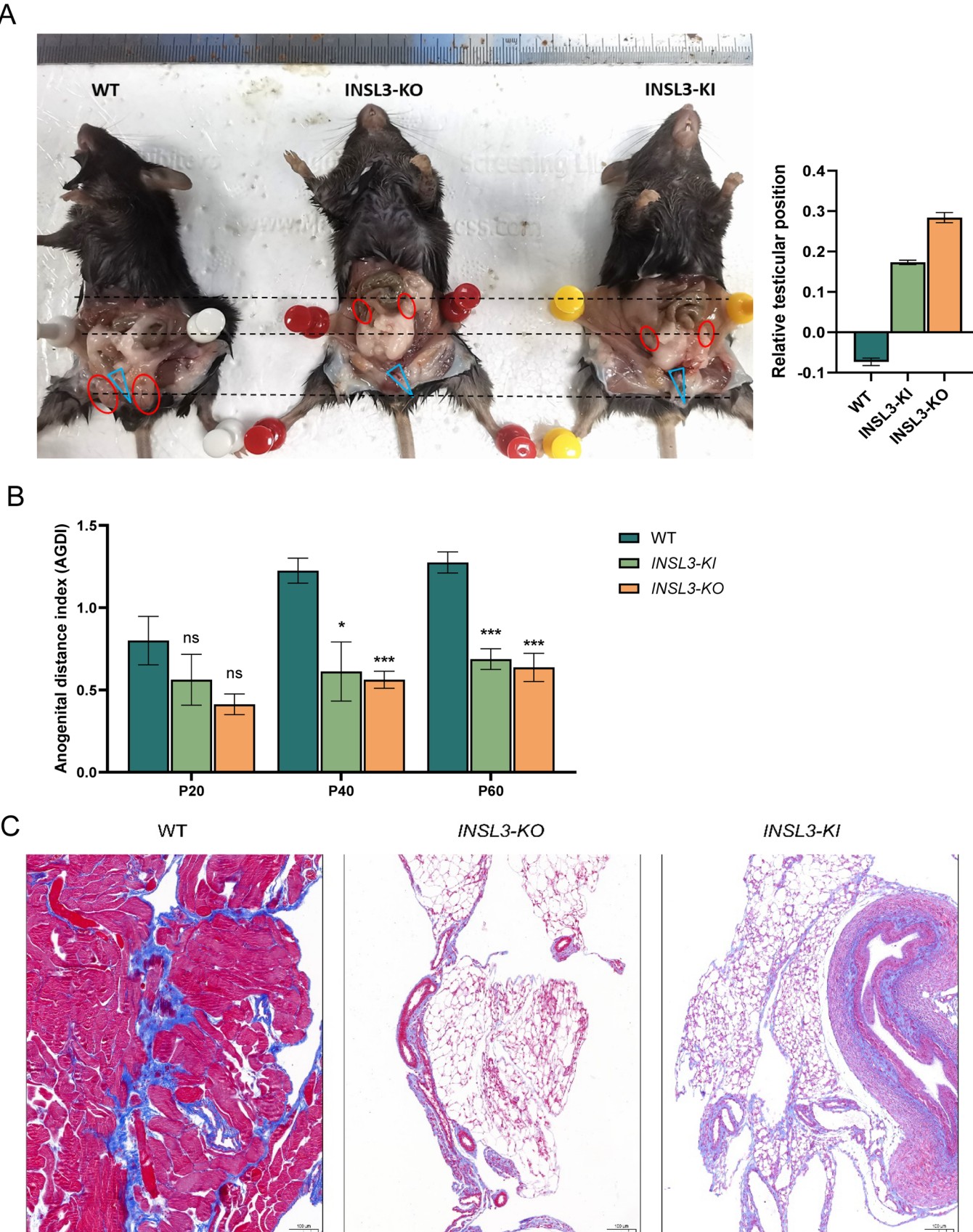

◄ **Figure 3. Cryptorchidism is present in cetacean *INSL3* gene knock-in mice.**

(A) Anatomical map showing the in situ testicular position in mice (red circle indicates testis, blue triangle indicates penis) with quantification based on the testis-to-penis position ratio (Data were shown as mean ± s.e.m. from $n = 8$ mice, biological replicates). (B) AGDI values of mice at different periods (Data were shown as mean ± s.e.m. from $n = 4$ mice, biological replicates. Statistical significance was determined using ANOVA tests. Exact p values: P20 WT vs KO, $p = 0.1464$; P20 WT vs KI, $p = 0.8039$; P20 KO vs KI, $p = 0.9261$; P40 WT vs KO, $p = 0.0009$; P40 WT vs KI, $p = 0.0444$; P40 KO vs KI, $p = 0.4160$; P60 WT vs KO, $p = 0.0008$; P60 WT vs KI, $p = 0.0005$; P60 KO vs KI, $p = 0.9978$. $*p < 0.05$; $***p < 0.005$; ns not significant). (C) Masson-stained micrograph of mouse gubernaculum at P15. Scale bar, 100 μm. Source data are available online for this figure.

cryptorchid KI mice. *INSL3*-KO mice were included at P8 and P23 to determine whether cetacean INSL3 simply recapitulates a KO phenotype; P40 *INSL3*-KO testes were not analyzed because prolonged cryptorchidism at this stage is dominated by nonspecific thermal damage. Across stages, DEG numbers indicated progressive damage in *INSL3*-KI testes, with *INSL3*-KO mice showing earlier and more extensive downregulation (Fig. 6A). At P8, *INSL3*-KI and *INSL3*-KO testes showed relatively few transcriptomic differences from WT, with modest suppression of genes linked to meiosis and spermatogenesis (Appendix Fig. S1). By P23, *INSL3*-KO testes displayed substantially more DEGs than *INSL3*-KI, and KO-specific downregulated genes were enriched for spermatogenesis-related functions, indicating more severe germ cell loss in *INSL3*-KO (Fig. 6B,C).

By P40, *INSL3*-KI testes exhibited 4,699 downregulated genes, predominantly involved in meiotic progression, sperm flagella formation, and motility, consistent with the absence of elongated sperm at this stage (Fig. 6D). Concurrently, upregulated genes were enriched in inflammatory pathways (including complement and coagulation cascades, TGF-β, and TNF signaling) and heat shock proteins, indicating a sustained inflammatory state likely triggered by elevated intra-abdominal temperature. These findings support a model in which cryptorchidism in *INSL3*-KI mice exposes the testes to chronic heat stress, leading to inflammation, meiotic disruption, and defective spermatid differentiation, ultimately resulting in infertility (Fig. 6E,F; Appendix Fig. S2). Interestingly, while naturally cryptorchid mammals maintain normal spermatogenesis despite abdominal testis position, the inflammation observed in *INSL3*-KI testes suggests that such species may exhibit distinct molecular adaptations, a possibility further explored in the Discussion.

# Discussion

Cryptorchidism is a common congenital defect in newborns. While early surgery can correct cryptorchidism, the risk of infertility and testicular cancer remains elevated in adulthood, with ~10% of males experiencing fertility issues having a history of the condition (Jungwirth et al, 2012). Recent research on cryptorchidism in mammals reveals molecular mechanisms involving accelerated evolution and convergence in genes and pathways such as Hedgehog signaling and WNT/BMP morphogenesis, contributing to testicular descent and thermal adaptation across species (Chai et al, 2021a; Chai et al, 2021b; Chai et al, 2022; Chai et al, 2021c).

In our evolutionary analysis of *INSL3* and *RXFP2*, we did not observe any premature stop codons or complete disruptions of known functional domains or conserved RXFP2-interacting motifs. Specifically, INSL3 comprises a signal peptide, B-chain, C-chain, and A-chain; the mature hormone contains only the A- and B-

chains, with the B-chain binding the high-affinity LRR region and the A-chain interacting with the TM domain of RXFP2 (Bruell et al, 2017; Esteban-Lopez and Agoulnik, 2020; Halls et al, 2005). In cetaceans, most amino acid changes are located in the signal peptide and C-chain—both removed during maturation—while the A- and B-chains are overall highly conserved, with only a few lineage-specific substitutions (e.g., a Valine residue in some rodents and cetaceans) (Fig. EV3). This substitution is not shared across all naturally cryptorchid taxa and is therefore unlikely to represent a convergent adaptation. This structural conservation supports the view that cetacean INSL3 likely retains partial functionality rather than being a fully inactivated pseudogene.

Intriguingly, we detected that the dN/dS ratios of *INSL3* and *RXFP2* in naturally cryptorchid species were significantly increased, though no convergent amino acid substitutions were identified. Additionally, we observed that the dN/dS ratio of *INSL3* in cetaceans was the highest among all cryptorchid groups (Table EV1). Based on these findings, we selected cetaceans as a representative cryptorchid lineage for functional validation by generating cetaceanized mice, analogous to humanized models, to characterize their functional changes in vivo. To achieve this, we precisely replaced the mouse *Insl3* coding sequence with the full-length cetacean *INSL3* coding sequence at the endogenous mouse locus, maintaining the native promoter and regulatory elements. This strategy ensures physiological expression levels and spatiotemporal patterns identical to wildtype mice, allowing us to directly assess the functional consequences of cetacean-specific sequence evolution in vivo. While this model cannot entirely disentangle the effects of cetacean ligand divergence from the absence of the mouse ligand, the tri-group design (WT, KI, KO) enables us to evaluate the degree to which cetacean INSL3 retains—or fails to retain—mouse INSL3 function.

RXFP2 serves as the sole receptor for INSL3 in vivo (Bogatcheva et al, 2003). Studies in RXFP2-transfected HEK293T cells have shown receptor activation leads to Gαs coupling, cAMP production, and subsequent CRE-dependent gene transcription (Halls et al, 2007). Our cellular experiments confirmed that cetacean INSL3 bound effectively to mouse RXFP2, likely due to conserved binding sites across species. The matching phenotypes in single and double knock-in mouse models further supported this (Fig. EV2). The primary reason for the insufficient activation of downstream pathways following INSL3 binding appeared to be the inadequate activation of PKA by cAMP. This may be due to INSL3 mediating other pathways to activate PKA in gubernacular cells, but this requires further investigation.

The gubernaculum plays a central role in testicular descent, originating in the inguinal region as mesenchymal cells within the oblique abdominal muscle fibers (Backhouse, 1964; Backhouse and Butler, 1960). In the first phase, INSL3 is recognized as the key hormone driving gubernacular thickening, while AMH and testosterone may also contribute (Kubota et al, 2002). The

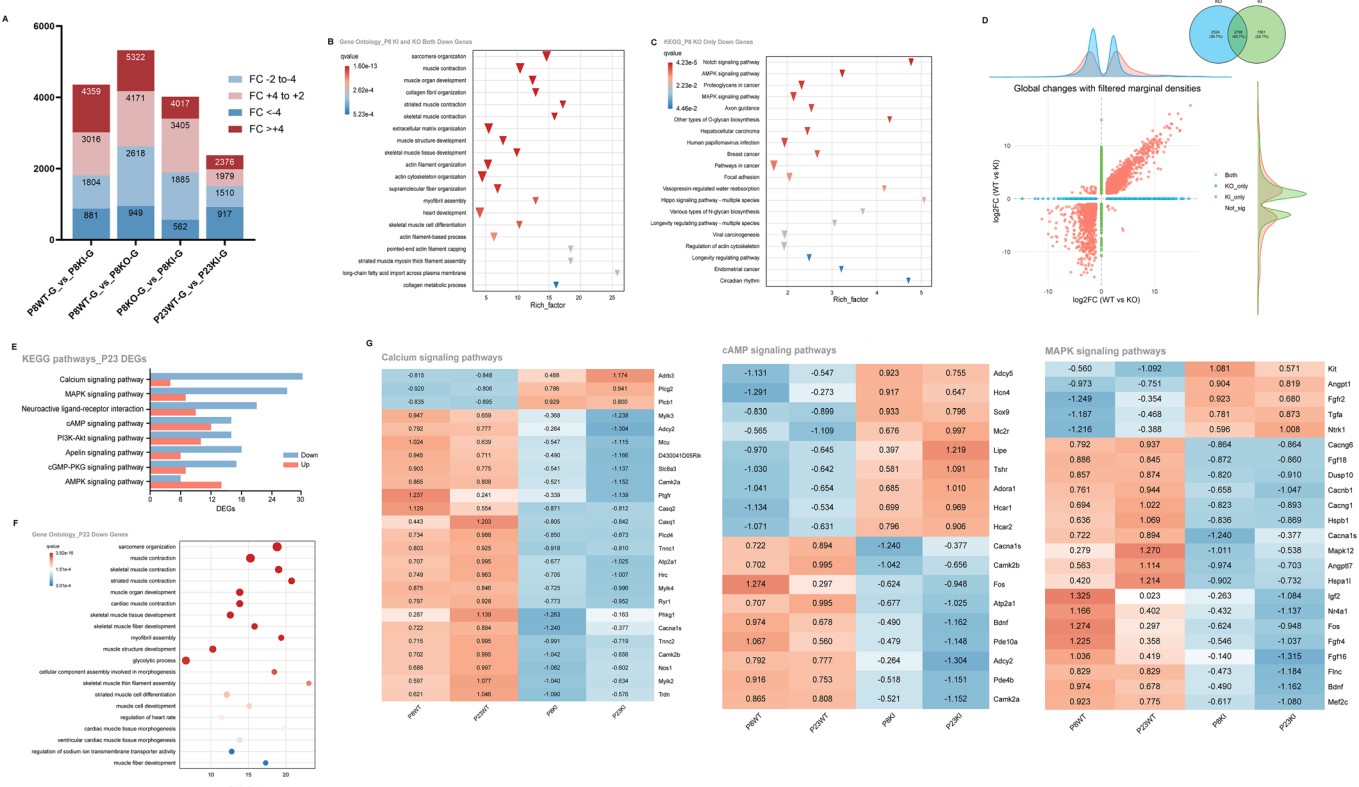

**Figure 4. Cetacea Insl3 induced changes in the gubernaculum transcriptome.**

(A) Number of differentially expressed genes in the gubernaculum of 8-day-old and 23-day-old mice. (B) Bubble map of GO terms enriched in genes downregulated in both *INSL3*-KI and *INSL3*-KO gubernaculum at P8 vs WT. (C) Bubble map of GO terms enriched in genes downregulated only in *INSL3*-KO gubernaculum at P8 vs WT. (D) Overview of transcriptomic changes in the gubernaculum at P8. Scatter plot of log₂FC values (WT vs *INSL3*-KO on the x-axis, WT vs *INSL3*-KI on the y-axis) with marginal density curves showing the distribution of each gene category (red: Both, orange: *INSL3*-KO_only, green: *INSL3*-KI_only, gray: Not_sig; *p*adj <0.01). The Venn diagram illustrates the overlap of significantly differentially expressed genes between the two comparisons (blue: WT vs *INSL3*-KO, green: WT vs *INSL3*-KI). Enrichment pathway analysis of differentially expressed genes in KGEE at P23 (E) stages. (F) Bubble map showing the downregulated GO terms enriched in *INSL3*-KI gubernaculum compared with the control at P23. (G) Heatmaps showing gene expression changes in the calcium, MAPK, and cAMP signaling pathways. RNA-seq data are based on *n* = 3 independent biological replicates per group. Differential expression was analyzed with DESeq2 (*p*adj < 0.01).

gubernaculum contacts the mesonephric duct and testis, forming a cord and bulb structure. At this stage, collagen fibers accumulate, and myoblast differentiation begins. During the second phase, the gubernaculum extended into the inguinal canal, forming the processus vaginalis. Testosterone, either directly or through genitofemoral nerve stimulation, induces gubernacular contraction, guiding the testis into the scrotum (Amann and Veeramachaneni, 2007). Following descent, the gubernaculum regressed, leaving a fibrous connection between the testis and scrotum, eventually forming part of the cremaster muscle (Backhouse, 1982). Through comparison of WT, *INSL3*-KO, and *INSL3*-KI mice, we found that in cetaceanized *INSL3*-KI mice, the first phase of descent occurred normally, but the second phase was abnormal. Histological analysis suggested impaired myoblast differentiation in these mice, preventing gubernacular contraction in the second phase of descent. Transcriptome sequencing of the *INSL3*-KI gubernaculum also revealed downregulation of genes enriched in muscle structure and function-related processes.

The molecular mechanisms underlying gubernaculum development remain poorly understood. Previous studies have shown that the INSL3/RXFP2 pathway induced WNT and BMP developmental

pathways, and in *Rxfp2⁻/⁻* mice, the expression of developmental signaling molecules such as β-catenin, NOTCH1, and WNT1 was reduced in the gubernaculum (De Toni et al, 2019; Esteban-Lopez and Agoulnik, 2020). Knockdown of β-catenin or Notch1 in the gubernaculum has also been shown to result in the absence of muscle layers (Kaftanovskaya et al, 2011). In our study, transcriptome analysis of the gubernaculum in *INSL3*-KI mice during pre- and post-scrotal stages revealed that in the post-scrotal stage (accompanied by cryptorchidism), a larger proportion of downregulated genes were observed, with GO enrichment indicating involvement in muscle structure and function. KEGG pathway enrichment showed that downregulated genes were concentrated in calcium signaling, MAPK, and cAMP pathways, suggesting that the mechanism of cryptorchidism in cetaceans may be due to INSL3 impairing the regulation of these pathways, leading to weakened gubernacular contraction during the second phase of testicular descent. Additionally, in the pre-scrotal stage, most differentially expressed genes were upregulated, with GO terms enriched for anterior-posterior patterning, calcium release, and extracellular matrix organization, suggesting a potential compensatory response to mild damage in the first phase of gubernacular development.

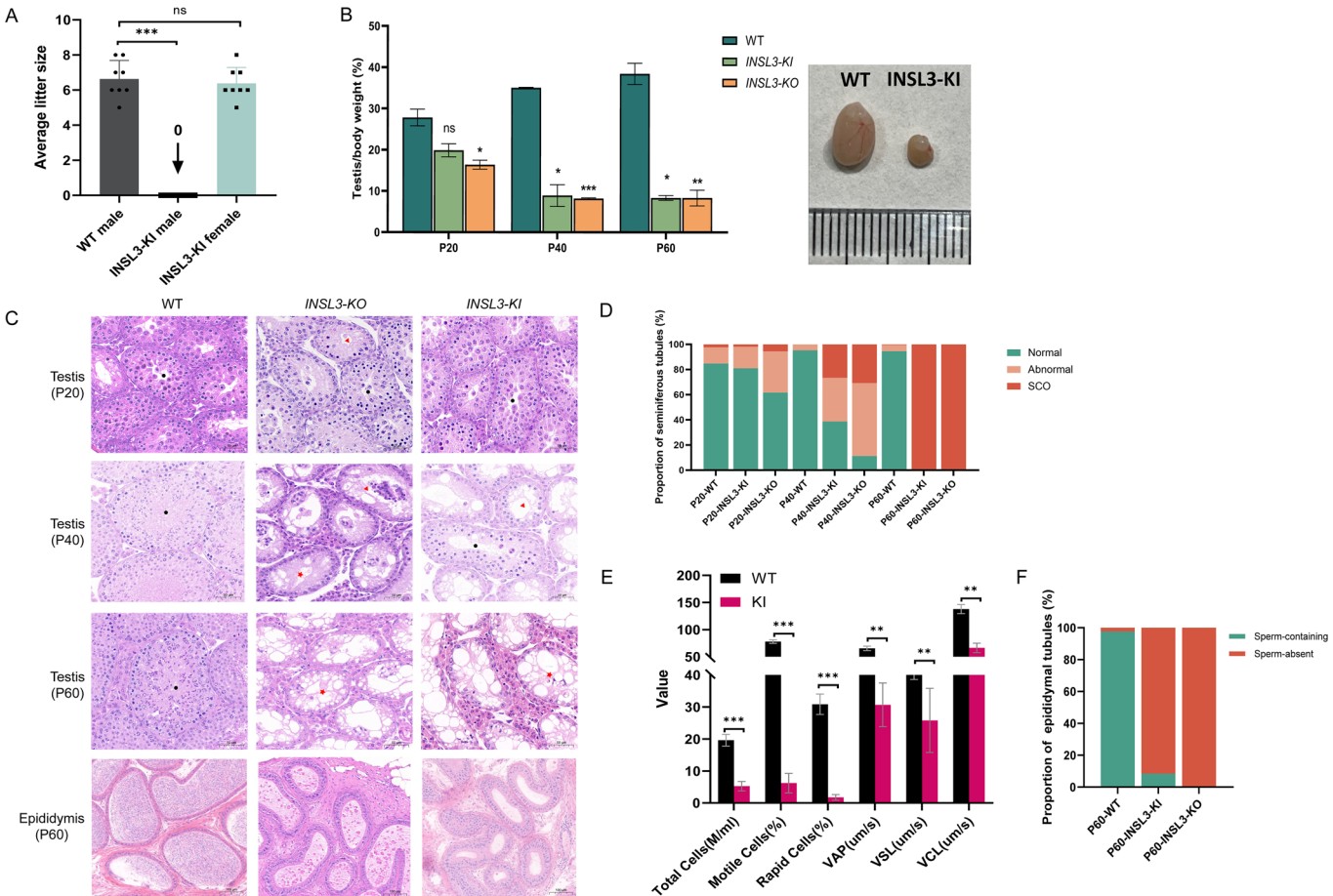

**Figure 5. Cryptorchidism in *INSL3*-KI mice impairs their fertility potential.**

(A) The average litter size was measured to evaluate the fertility of mice (Data were shown as mean ± s.e.m. $n = 8$ mice, biological replicates. Statistical significance was determined using two-tailed $t$-tests. Exact $p$ values: WT male vs KI male, $p = 0.0009$; WT male vs KI female, $p = 0.6217$. ***$p < 0.001$; ns not significant). (B) Testicular organ index of mice at different periods (Data were shown as mean ± s.e.m. $n = 4$ mice, biological replicates. Statistical significance was determined using ANOVA tests. Exact $p$ values: P20 WT vs KO, $p = 0.0434$; P20 WT vs KI, $p = 0.1054$; P20 KO vs KI, $p = 0.4562$; P40 WT vs KO, $p = 0.0009$; P40 WT vs KI, $p = 0.0261$; P40 KO vs KI, $p = 0.7598$; P60 WT vs KO, $p = 0.0018$; P60 WT vs KI, $p = 0.0198$; P60 KO vs KI, $p > 0.9999$. *$p < 0.05$; **$p < 0.01$; ***$p < 0.001$; ns, not significant). (C) HE-stained micrograph of mouse testis and epididymis. Scale bar, 100 μm. Black circles indicate normal seminiferous tubules (all germ cell layers present and correctly positioned), red triangles indicate abnormal tubules (germ cell loss or misplacement), and red stars indicate Sertoli cell-only (SCO) tubules (no germ cells). Scale bar, teste: 50 μm, epididymis: 100 μm. (D) Percentage stacked bar chart showing the proportion of seminiferous tubules classified as normal, abnormal, or Sertoli cell-only (SCO) in WT, *INSL3*-KO, and *INSL3*-KI mice at different developmental stages. Data were obtained from 3–4 mice per group, with no fewer than 150 tubules evaluated per group. (E) ACASA system was used to measure spermatozoa motility parameters. Spermatozoa were collected from the cauda epididymis of P60 mice. VAP average path velocity, VSL straight-line velocity, VCL curvilinear velocity (Data were shown as mean ± s.e.m. $n = 5$ mice, biological replicates. Statistical significance was determined using two-tailed $t$-tests. Exact $p$ values: WT vs KI, Total Cells $p = 0.0009$, Motile Cells $p < 0.0001$, Rapid Cells $p = 0.0001$, VAP $p = 0.0046$, VSL $p = 0.0098$, VCL $p = 0.001$. *$p < 0.05$, **$p < 0.01$, ***$p < 0.001$). (F) Percentage stacked bar chart showing the proportion of epididymal tubules containing or lacking sperm in WT, *INSL3*-KO, and *INSL3*-KI mice at P60. Data were obtained from four mice per group, with no fewer than 50 tubules evaluated per group. Source data are available online for this figure.

A recent study identified a connection between cryptorchidism and a higher incidence of male infertility (Trussell and Lee, 2004). An observational study highlighted that 89% of untreated individuals with bilateral cryptorchidism were diagnosed with azoospermia (Hadziselimovic, 2002). The cetacean *INSL3*-KI mice we developed exhibited both bilateral cryptorchidism and male infertility, characterized by seminiferous tubule atrophy, progressive loss of germ cells, and an inability to produce mature sperm (Fig. 5C). This contrasted with cetaceans' ability to maintain normal reproduction despite natural cryptorchidism, suggesting that other key genes related with male reproduction function must have evolved for a healthy cryptorchidism in cetaceans. Analysis of

the testicular transcriptome in *INSL3*-KI mice across different stages further revealed that post-cryptorchidism testicular damage is progressive. A key factor likely contributing to infertility in cryptorchid *INSL3*-KI mice is the abnormally elevated inflammatory response in the testes during the first wave of postnatal spermatogenesis. Undescended testes are exposed to elevated intra-abdominal temperatures, which can trigger heat stress responses, promote inflammatory cascades, and impair meiotic progression (Aldahhan et al, 2021; Sun et al, 2020; Wang et al, 2024). Consistent with this, our RNA-seq analysis revealed coordinated upregulation of heat stress–responsive genes (e.g., *Hspa1b* and *Hsp90ab1*) and inflammatory mediators, alongside downregulation of key meiotic

regulators (e.g., *Rec8* and *Cpeb3*) in *INSL3*-KI testes. These transcriptional changes suggest that heat-induced inflammation may interfere with meiotic entry or progression, leading to germ cell loss and defective spermatogenesis. While we did not directly assess inflammation in naturally cryptorchid species, their ability to sustain normal spermatogenesis despite abdominal testis position raises the possibility that they have evolved mechanisms to modulate or suppress testicular inflammation, thereby protecting germ cell development. Elucidating such mechanisms will be an important direction for future research.

# Methods

### Reagents and tools table

| Reagent/resource | Reference or source | Identifier or catalog number |
| --- | --- | --- |
| **Experimental models** | | |
| HEK293T cells (*H. sapiens*) | Procell | CL-0005 |
| Primary gubernacular cells (*M. musculus*) | C57BL/6 strain, male, postnatal day 5 | This study |
| C57BL/6J (*M. musculus*) | GemPharmatech Co., Ltd. | This study |
| **Recombinant DNA** | | |
| pcDNA3.1(+) | Thermo Fisher | V79020 |
| **Antibodies** | | |
| Rabbit anti-INSL3 | ABclonal | Catalog NO. A5728 |
| Rabbit Flag-Tag Ab | Affinity | Cat. # T0003 |
| Rabbit anti-CREB(phosphor S133) antibody [E113] | Abcam | ab32096 |
| Rabbit beta-tubulin | ABclonal | A12289 |
| **Oligonucleotides and other sequence-based reagents** | | |
| PCR Primers | This study | Appendix Table S1 |
| **Chemicals, enzymes, and other reagents** | | |
| DMEM high glucose | WISENT | 319-005 |
| Fetal bovine serum (FBS) | WISENT | 086-150/110 |
| TRYPSIN 0.25%/2.21 mM EDTA in HBSS | WISENT | 325-043 |
| Collagenase, Type 1 | Diamond | A004194-0001 |
| Actin-Tracker Green-488 | Beyotime | C2201S |
| Multiwell TC Treated Plates, 6-Well, TC Treated | BBI | F603201-0001 |
| recombinant proteins | Bankpeptide, Hefei, China | Customization |
| CCK-8 Cell Counting Kit | Vazyme | A311-01 |
| Mouse Cyclic Adenosine Monophosphate (cAMP) ELISA Kit | JONLNBIO | JL13362 |
| Mouse Protein kinase A(PKA) ELISA Kit | JONLNBIO | JL20309 |
| RIPA Lysis Buffer | Beyotime | P0013C |

| Reagent/resource | Reference or source | Identifier or catalog number |
| --- | --- | --- |
| 180 kDa Plus Prestained Protein Marker | Vazyme | MP201 |
| 2 × Rapid Taq Master Mix | Vazyme | P222 |
| VeZol Reagent | Vazyme | R411 |
| Hifair III first Strand cDNA Synthesis SuperMix qPCR | YEASEN | 11141ES10 |
| Bouin's fixative solution | Servicebio | G1121 |
| Modified HTF Medium | FUJIFILM Irvine Scientific | 90126 |
| **Software** | | |
| GraphPad Prism 10 | https://www.graphpad.com | |
| ImageJ | https://imagej.nih.gov/ij/index.html | |
| CaseViewer | https://www.3dhistech.com/news/caseviewer-becomes-slideviewer/ | |
| R (4.2.2) | https://www.r-project.org/ | |
| RStudio (2022.07.2) | https://posit.co/download/rstudio-desktop/ | |
| **Other** | | |
| Illumina novaseq,6000 | Illumina | |

## Ethics statement

All the animal experiments were carried out in strict accordance with recommendations in the Regulations on the Management of Laboratory Animals of Nanjing Normal University. Protocols were approved by the Animal Ethical and Welfare Committee of Nanjing Normal University (approval ID: IACUC-20230402). All possible efforts were employed to reduce the number of animals used and also to minimize animal suffering.

## Animals

Cetacean *INSL3* gene knock-in mice in the C57BL/6 J background were generated by the GemPharmatech Co., Ltd. (Nanjing, China). The minke whale (*Balaenoptera acutorostrata*) *INSL3* allele was generated by targeting the cDNA of minke whale *INSL3* to its locus using homologous recombination. Both polymerase chain reaction (PCR) and Southern blotting approaches were used to verify the targeting to the *INSL3* locus. Targeted embryonic stem cell clones were used to produce chimeric animals using eight-cell stage injection. The knock-in male mice carrying a neomycin cassette were mated with Cre females to obtain offspring, then inbred to produce *INSL3*-KI homozygous and WT littermates. *INSL3*-KI homozygous mice without the neo cassette were used in all experiments. WT C57BL/6J mice, aged 10–14 weeks, were used in control groups. *RXFP2*-KI homozygotes were obtained using the same strategy, and *INSL3/RXFP2*-DKI mice were generated by breeding these with *INSL3*-KI mice. Animals were genotyped using primer sequences as shown in Appendix Table S1. The exclusive expression of cetacean INSL3 protein in these mice was confirmed by RT-PCR and Western blotting (Fig. EV4). Mice were housed in a conventional facility at 21 °C on a 12 h light–dark cycle with unrestricted access to food and water.

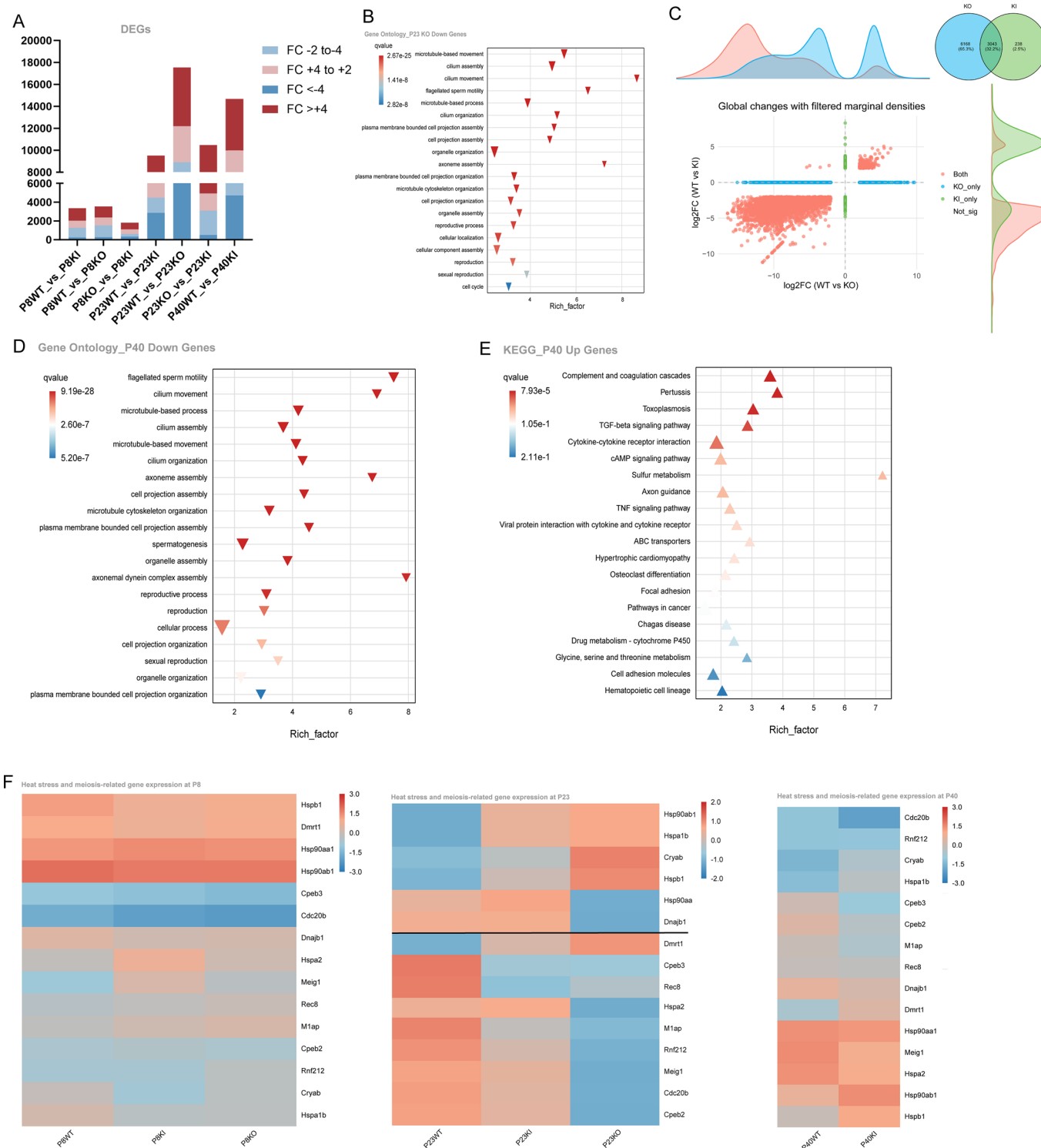

**Figure 6.  Transcriptome differences between *INSL3*-KI and wild-type testes.**

(**A**) Number of differentially expressed genes in the testis of 8, 23, and 40-day-old mice. (**B**) Bubble map showing the downregulated GO terms enriched in *INSL3*-KO testis compared with control at P23. (**C**) Overview of transcriptomic changes in the testes at P23. Scatter plot of log₂FC values (WT vs *INSL3*-KO on the x-axis, WT vs *INSL3*-KI on the y-axis) with marginal density curves showing the distribution of each gene category (red: Both, orange: *INSL3*-KO_only, green: *INSL3*-KI_only, gray: Not_sig; padj < 0.01). The Venn diagram illustrates the overlap of significantly differentially expressed genes between the two comparisons (blue: WT vs *INSL3*-KO, green: WT vs *INSL3*-KI). (**D**) Bubble map showing the downregulated GO terms enriched in *INSL3*-KI testis compared with control at P40. (**E**) Pathway enrichment analysis of upregulated genes in KGEE at the P40 stage. (**F**) Heatmaps showing gene expression changes in the heat stress and meiosis-related genes. RNA-seq data are based on $n = 3$ independent biological replicates per group.

## Species coverage and sequence acquisition

The evolutionary analyses covered a total of 46 mammals (include species with natural 'cryptorchidism' state and their close relatives) from 12 orders: Cetartiodactyla (sperm whale *Physeter catodon*, baiji *Lipotes vexillifer*, white whale *Delphinapterus leucas*, Yangtze finless porpoise *Neophocaena asiaeorientalis*, Indo-Pacific humpback dolphin *Sousa chinensis*, bottlenose dolphin *Tursiops truncatus*, minke whale *Balaenoptera acutorostrata*, blue whale *Balaenoptera musculus*, hippopotamus *Hippopotamus amphibius*, wild Bactrian camel *Camelus ferus*, cattle *Bos taurus*, goat *Capra hircus*, sheep *Ovis aries* and wild boar *Sus scrofa*), Perissodactyla (horse *Equus caballus* and white rhinoceros *Ceratotherium simum*), Carnivora (cat *Felis catus*, wolf *Canis lupus*, European polecat *Mustela putorius*, walrus *Odobenus rosmarus*, Weddell seal *Leptonychotes weddellii* and giant panda *Ailuropoda melanoleuca*), Chiroptera (black flying fox *Pteropus alecto*, large flying fox *Pteropus vampyrus* and David's myotis *Myotis davidii*), Rodentia (European rabbit *Oryctolagus cuniculus*, prairie vole *Microtus ochrogaster*, Chinese hamster *Cricetulus griseus*, brown rat *Rattus norvegicus*, house mouse *Mus musculus*, naked mole-rat *Heterocephalus glaber*, common degu *Octodon degus*, long-tailed chinchilla *Chinchilla lanigera* and guinea pig *Cavia porcellus*), Primates (northern white-cheeked gibbon *Nomascus leucogenys*, humans *Homo sapiens*, rhesus macaque *Macaca mulatta* and treeshrew *Tupaia chinensis*), Eulipotyphla (common shrew *Sorex araneus*, European hedgehog *Erinaceus europaeus* and star-nosed mole *Condylura cristata*), Didelphimorphia (gray short-tailed opossum *Monodelphis domestica*), Proboscidea (African bush elephant *Loxodonta africana*), Sirenia (West Indian manatee *Trichechus manatus*), Tubulidentata (aardvarks *Orycteropus afer*), and Ornithorhynchidae (platypus *Ornithorhynchus anatinus*). The testicular position of the above mammals was collected from previous studies (Chai et al, 2021a; Hutson et al, 2016).

The protein-coding sequences (CDSs) were then downloaded from the NCBI database (https://www.ncbi.nlm.nih.gov/). In addition, partial or unannotated CDS was further verified using Blastn searches with custom Perl scripts. The longest transcript was retained for each gene in this analysis. To obtain better quality sequence alignments, we performed multiple sequence alignments for each orthologous gene using PRANK v.170427 (Löytynoja, 2014) combined with MACSE v2 (Ranwez et al, 2018) in the codon mode. The aligned sequences were then trimmed using Gblocks v0.91 (Talavera and Castresana, 2007) with default settings.

## Molecular evolution analyses

To assess the selective pressure on genes, we estimated the ratio of nonsynonymous (dN) and synonymous (dS) substitution rates (dN/dS), commonly referred to as the $\omega$ ratio, implemented in the CodeML program of the PAML software package v4.9 (Yang, 2007). The phylogenetic tree of the representative species used here was obtained from the TimeTree (Kumar et al, 2022). Terminal branches or taxa of all naturally cryptorchidism species were labeled as foreground branches and other branches as background branches. We detected the relaxed selection of genes using the branch model. The one-ratio model assumes that all branches on the phylogenetic tree have the same $\omega$ ratio against an alternative hypothesis (two-ratio model), which allows the $\omega$ ratio of the foreground branch to differ from that of the background branch.

We used RELAX (DataMonkey, http://www.datamonkey.org/ (Wertheim et al, 2015) to detect selective pressure relaxation in naturally cryptorchid species, labeling their terminal branches as test branches and others as references. Then, we executed a likelihood ratio test with a chi-square distribution and applied the false discovery rate correction for multiple testing.

## Primary cell culture

The gubernaculum from 3- to 5-day-old mice were removed under an operating magnifier and incubated in Dulbecco's modified Eagle's medium (DMEM) with 1 mg/ml type I collagenase for one hour. Cells were collected by centrifugation and cultured in DMEM containing 10% fetal bovine serum (FBS) under 5% $CO_2$ and 95% air. After 1 week, primary cells were harvested by trypsinization and transferred into each well of a six-well culture plate (Zhang et al, 2012). Subcultured cells were randomly grouped to apply concentration gradients of mouse and minke whale INSL3 protein cultures, respectively. All recombinant proteins were obtained from chemical synthesis (Bankpeptide, Hefei, China). Five independent experiments were performed, each of which was from a new primary culture preparation.

Primary gubernacular cells were isolated from *Mus musculus* (C57BL/6 strain, male, postnatal day 5). No genetic modification was introduced. Primary cultures were confirmed by morphology and immunostaining for $\alpha$-smooth muscle actin. Mycoplasma testing was performed routinely (negative).

## Cell proliferation assay

1. Gubernaculum cells were seeded into 96-well plates and cultured in DMEM supplemented with 10% FBS for 24 h.
2. Cells were treated as indicated and harvested at the designated time points.
3. Cell proliferation was measured using the Cell Counting Kit-8 (CCK-8; Vazyme, Nanjing, China) according to the manufacturer's instructions.
4. Absorbance at 450 nm was recorded using a microplate reader (Molecular Devices, Sunnyvale, CA, USA).

## Immunofluorescence staining

1. Gubernaculum cell monolayers were washed with phosphate-buffered saline (PBS) and fixed in 4% formaldehyde/PBS for 20 min.
2. Cells were permeabilized with 0.2% Triton X-100/PBS for 10 min.
3. F-actin was stained using Actin-Tracker Green (C2201S, Beyotime, China) diluted in PBS, incubated for 40 min.
4. Coverslips were mounted in DTG mounting medium (Sigma, USA).
5. Fluorescence images were acquired using a Leica fluorescence microscope.

## ELISA for cAMP and PKA

1. cAMP and PKA levels were quantified using commercial ELISA kits (Jianglai Biology, Shanghai, China).

2. Fifty microliters of the sample were incubated with 100 μL HRP-conjugated reagent for 1 h at 37 °C.
3. After washing, 50 μL detection reagent A and 50 μL detection reagent B were added sequentially and incubated for 15 min at 37 °C.
4. The reaction was terminated by adding 50 μL stop solution.
5. Absorbance at 450 nm was measured using an ELISA analyzer.

## Western blotting

1. Testes or cell lysates were prepared in RIPA buffer supplemented with freshly added protease inhibitor cocktail (Beyotime, China).
2. Proteins were separated on 8 or 10% SDS–PAGE gels and transferred to PVDF membranes.
3. Membranes were blocked with 5% BSA in TBST, incubated overnight at 4 °C with primary antibodies (ABclonal, China), washed three times in TBST, and incubated with HRP-conjugated secondary antibodies for 1 h at room temperature.
4. Protein sizes were verified using EZ-Run Prestained Rec Protein Ladder (Vazyme, Nanjing, China).
5. Signals were visualized using an Azure Imaging System 600, and figures were prepared in Adobe Photoshop.

## RT-qPCR

1. Total RNA was extracted from testes using Trizol Reagent (Vazyme, Nanjing, China) following the manufacturer's instructions.
2. Two micrograms of RNA were reverse-transcribed using Hifair® I First Strand cDNA Synthesis SuperMix for qPCR (YEASEN, Shanghai, China).
3. Quantitative PCR was performed with SYBR Green Mix (YEASEN) on a LightCycler 480 II system (Roche Applied Science, Mannheim, Germany).
4. Relative mRNA expression was determined by the ΔΔCt method.
5. Primer sequences are provided in Appendix Table S1.

## Histomorphological analysis

1. Freshly dissected testes were fixed in Bouin's solution (G1121, Servicebio, China) for 20 h at room temperature and then transferred to 70% ethanol for storage before processing.
2. Samples were dehydrated through graded ethanol (70, 80, 95, and 100%), cleared in xylene, and embedded in paraffin.
3. Paraffin blocks were sectioned at 5 μm using a rotary microtome (Leica, Germany). Sections were mounted on glass slides and dried overnight at 37 °C.
4. For H&E staining, sections were deparaffinized, rehydrated, stained in hematoxylin for 5–10 min, rinsed, differentiated in acid alcohol, counterstained in eosin for 1–3 min, dehydrated, cleared, and mounted.
5. For gubernaculum analysis, tissues were fixed in 4% paraformaldehyde for 20 h before paraffin embedding, and sections were stained with Masson's trichrome (Servicebio, China) according to the manufacturer's protocol. Collagen fibers were visualized in blue and muscle fibers in red.

6. Stained sections were examined with a Leica DM5000 bright-field microscope, and images were captured using Leica Application Suite software.

## Fertility studies

1. Adult *INSL3*-KI males and WT females were housed together for 6 months in a standard breeding protocol (1 male:2 females per cage).
2. Litter size and number were recorded at each parturition to assess reproductive performance.
3. After 6 months of continuous breeding, male mice were euthanized under approved protocols.
4. Testes, epididymides, and gubernaculum were collected, weighed, photographed, and preserved for further analysis.

## Computer-assisted sperm analysis (CASA)

1. Spermatozoa were released from the cauda epididymis into modified HTF medium containing 10% FBS.
2. Samples were incubated for 10 min at 37 °C under 5% $CO_2$ to allow sperm to disperse.
3. A 10 μL suspension was loaded into macroslides and immediately analyzed using the CASA system (IVOS II; Hamilton Thorne, USA).
4. The system quantified sperm motility (%) and measured motion parameters, including VAP (average path velocity), VSL (straight-line velocity), VCL (curvilinear velocity), ALH (amplitude of lateral head displacement), BCF (beat cross frequency), STR (straightness of track), and LIN (linearity of track).

## RNA-seq analysis

Total RNA was extracted using TRIzol Reagent (Life Technologies, CA, USA), and RNA concentration and purity were measured with a NanoDrop 2000 (Thermo Fisher Scientific, DE, USA). Sequencing libraries were prepared using the Hieff NGS Ultima Dual-mode mRNA Library Prep Kit for Illumina (Yeasen Biotechnology, Shanghai) according to the manufacturer's protocol, with index codes added for sample identification. Libraries were sequenced on an Illumina NovaSeq platform to generate 150 bp paired-end reads. The raw reads were further processed with a bioinformatic pipeline tool, the BMKCloud (www.biocloud.net) online platform. Raw data (raw reads) of fastq format were first processed through in-house Perl scripts. Hisat2 tools soft were used to map with the reference genome. Differential expression analysis of the two groups was performed using DESeq2. The resulting *P* values were adjusted using Benjamini and Hochberg's approach for controlling the false discovery rate. Genes with an adjusted *P* value < 0.01 and fold change ≥ 2 found by DESeq2 were assigned as differentially expressed. Gene Ontology (GO) enrichment analysis of the differentially expressed genes (DEGs) was implemented by the clusterProfiler packages based on Wallenius non-central hyper-geometric distribution, which can adjust for gene length bias in DEGs. KEGG (Kanehisa et al, 2008) is a database for understanding biological functions from cells to ecosystems using molecular data; we used the KOBAS database (Mao et al, 2005) and

clusterProfiler to test the enrichment of differentially expressed genes in KEGG pathways.

## Statistics

All statistical analyses were performed using R software (v4.3.2). Prior to applying statistical tests, data distribution was assessed using the Shapiro–Wilk test. For normally distributed data, parametric tests were applied: Student's *t*-test (two groups) or one-way analysis of variance (ANOVA, multiple groups). For non-normally distributed data, nonparametric Wilcoxon–Mann–Whitney tests were used. When ANOVA indicated significant differences, post hoc Tukey's multiple comparison tests were performed.

Sample sizes were chosen based on standard practice in reproductive biology and previous studies of comparable design; no formal sample size calculation was performed. Variance homogeneity between groups was examined using Levene's test, and assumptions of the applied statistical tests were met unless otherwise indicated. Data were presented as mean ± SEM unless stated otherwise, and individual data points are shown where applicable. A *P* value <0.05 was considered statistically significant.

## Adherence to community standards

This study adheres to relevant community standards. All animal experiments were reported in accordance with the ARRIVE guidelines. RNA-seq experiments comply with the MINSEQE reporting standards, and sequencing data have been deposited in a public repository with accession numbers provided in the manuscript. We also followed the general recommendations of the ICMJE for scientific reporting. No clinical trials, systematic reviews, or prognostic marker studies were conducted; thus, CONSORT, PRISMA, and REMARK guidelines are not applicable.

## Data availability

Raw RNA-seq data generated in this study have been deposited in the NCBI Gene Expression Omnibus (GEO) under accession number GSE305702 (URL: https://www.ncbi.nlm.nih.gov/geo/query/acc.cgi?acc=GSE305702).

The source data of this paper are collected in the following database record: biostudies:S-SCDT-10_1038-S44319-025-00636-w.

## Peer review information

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

## Acknowledgements

We thank Huiling Li of the Sha Jiahao Research Group at Nanjing Medical University for help in testing sperm motility. This work was supported by the National Natural Science Foundation of China (grant nos. 32030011, U24A20362, 32270442, and 32070409), National Key Program of Research and Development, Ministry of Science and Technology of China (grant no. 2022YFF1301600), PI Project of Southern Marine Science and Engineering Guangdong Laboratory (Guangzhou) (GML2021 GD0805), and the Priority Academic Program Development of Jiangsu Higher Education Institutions.

## Author contributions

**Yu Zheng**: Conceptualization; Data curation; Formal analysis; Visualization; Writing—original draft. **Simin Chai**: Resources; Software; Supervision; Investigation; Methodology; Writing—review and editing. **Cuijuan Zhong**: Data curation; Formal analysis; Visualization; Methodology. **Yixuan Sun**: Conceptualization; Data curation; Supervision; Validation. **Shixia Xu**: Supervision; Funding acquisition; Validation; Project administration. **Wenhua Ren**: Conceptualization; Supervision; Funding acquisition; Validation. **Guang Yang**: Conceptualization; Supervision; Funding acquisition; Validation; Writing—original draft; Project administration; Writing—review and editing.

Source data underlying figure panels in this paper may have individual authorship assigned. Where available, figure panel/source data authorship is listed in the following database record: biostudies:S-SCDT-10_1038-S44319-025-00636-w.

## Disclosure and competing interests statement

The authors declare no competing interests.

# Expanded View Figures

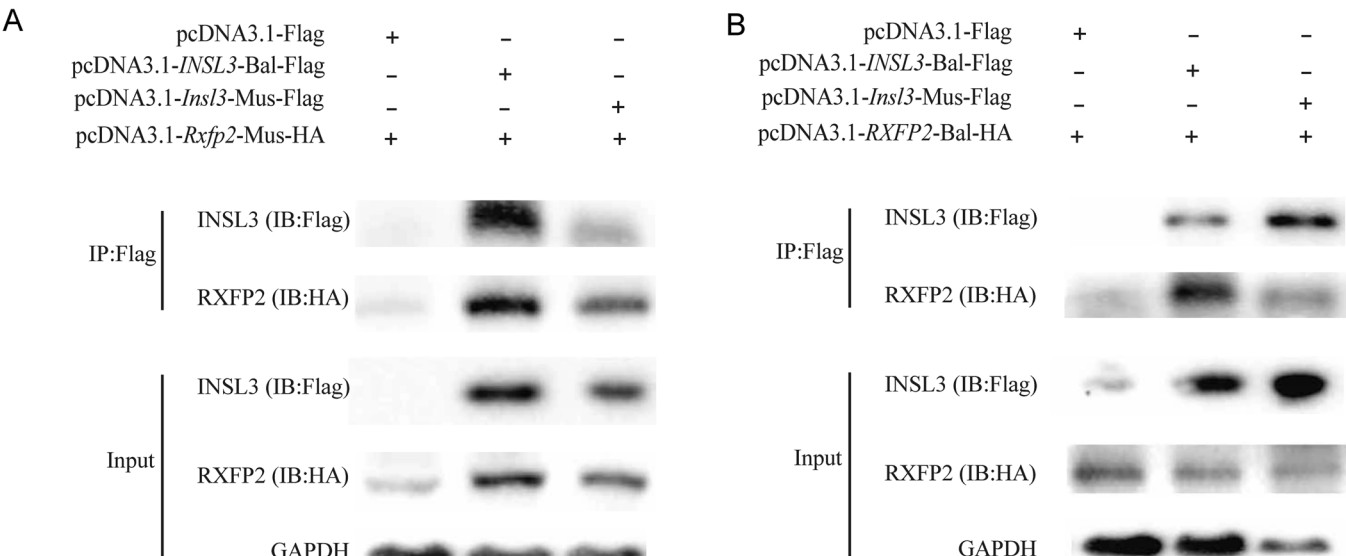

**Figure EV1. Co-IP assay of INSL3 and RXFP2 binding capacity in cetaces and mice.**

(A) Binding ability of the mouse RXFP2 receptor to mouse INSL3 and cetacean INSL3. (B) The binding ability of the cetacean RXFP2 receptor to mouse INSL3 and cetacean INSL3, respectively. Data were representative of $n = 3$ independent experiments.

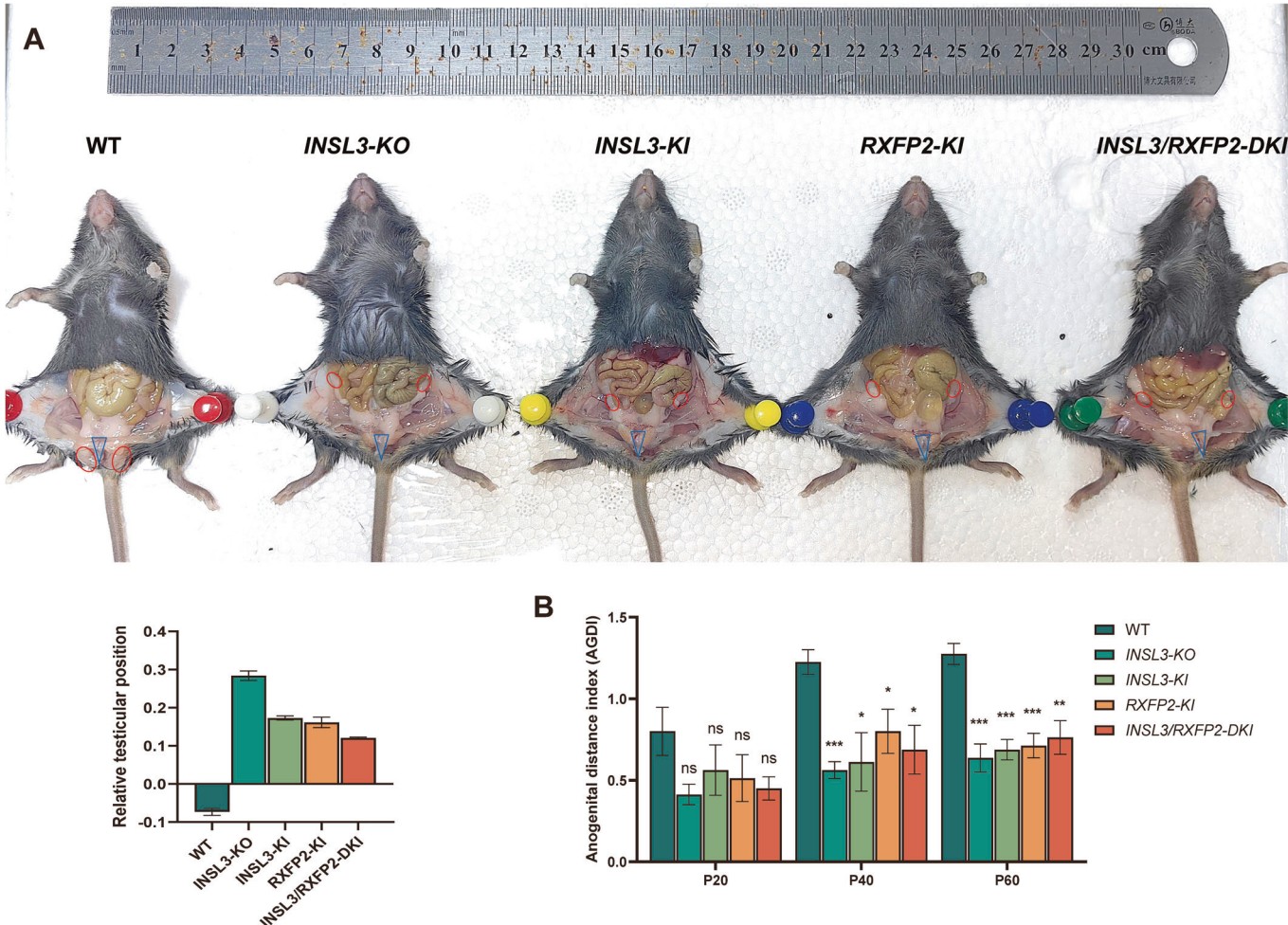

**Figure EV2. Cryptorchidism is present in three cetaceanized mice.**

(A) Anatomical map showing the in situ testicular position in mice (red circle indicates testis, blue triangle indicates penis) with quantification based on the testis-to-penis position ratio (Data were shown as mean ± s.e.m. from $n = 8$ mice, biological replicates). (B) AGDI values of mice at different periods (Data are shown as mean ± s.e.m. from $n = 4$ mice, biological replicates. Statistical significance was determined using ANOVA tests. Exact $p$ values: P40 WT vs KO, $p = 0.0009$; P40 WT vs *INSL3*-KI, $p = 0.0444$; P40 WT vs *RXFP2*-KI, $p = 0.0496$; P40 WT vs DKI, $p = 0.0383$; P60 WT vs KO, $p = 0.0008$; P60 WT vs *INSL3*-KI, $p = 0.0005$; P60 WT vs *RXFP2*-KI, $p = 0.0011$; P60 WT vs DKI, $p = 0.0110$. *$p < 0.05$; **$p < 0.01$***$p < 0.005$; ns not significant).

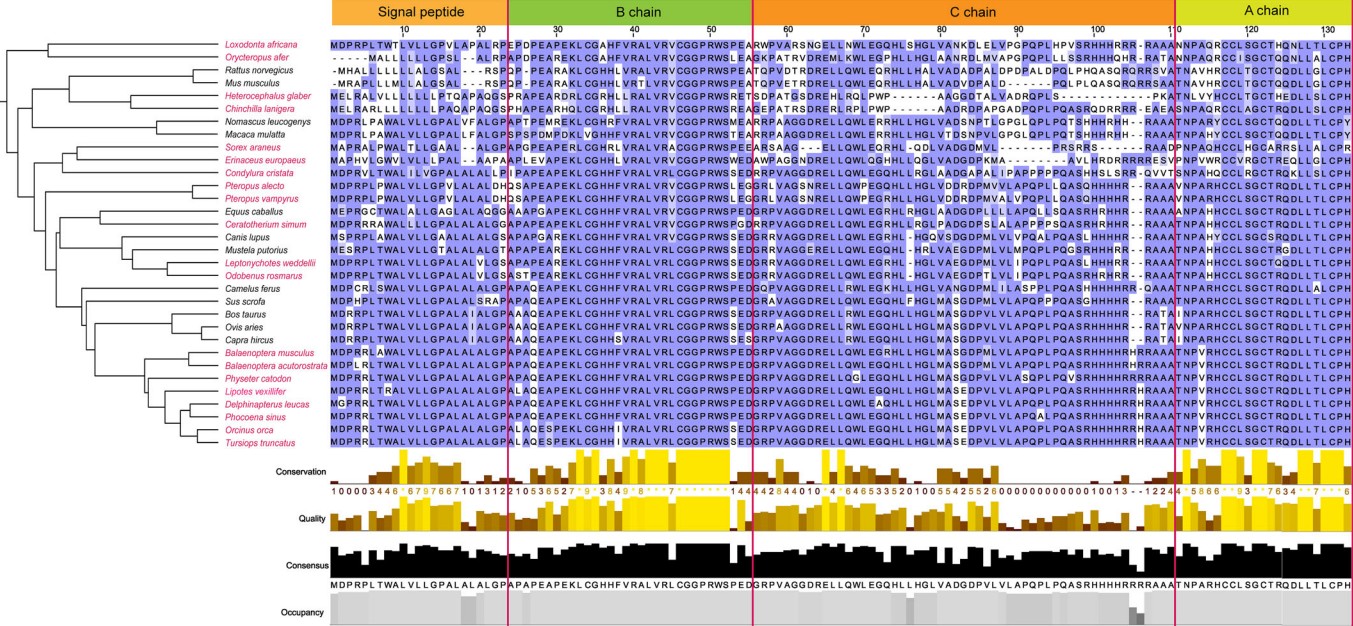

**Figure EV3. The sequence of mammalian INSL3.**

The red Latin name is cryptorchid mammals, and the black Latin name is scrotal mammals. The two functional domains of the mature INSL3 protein are marked in red boxes.

**Figure EV4. The cetacean *INSL3* gene knock-in mice were successfully constructed.**

(A) Agarose gel electrophoresis plot of genotyping. Lane 1: Maker DL2000; Lane 1, 4, 7, 9, 10, 13: heterozygote; Lane 2, 3, 14: homozygote; Lane 5, 6, 8, 11, 12: wildtype; Lane 15: blank control. (B) Agarose gel electrophoresis plot of RT-PCR. Lane 1: Maker DL2000; Lane 2–8: the primer was mouse *INSL3*; Lane 9–15: the primer was cetacean *INSL3*. (C) WB of mouse testicular proteins. Data were representative of $n = 3$ independent experiments.

