## [Peer Review File · EMBO Reports]

Evolutionary Relaxation and Functional Change of INSL3 and RXFP2 May Underlie Natural Cryptorchidism in Mammals

Yu Zheng, Simin Chai, Cuijuan Zhong, Yixuan Sun, Shixia Xu, Wenhua Ren, and Guang Yang

Corresponding author(s): Guang Yang (gyang@nynu.edu.cn)

Review Timeline:

Submission Date:	9th Mar 25
Editorial Decision:	20th May 25
Revision Received:	19th Aug 25
Editorial Decision:	14th Oct 25
Revision Received:	18th Oct 25
Accepted:	28th Oct 25

Editor: Yehu Moran

Transaction Report:

Dear Prof. Yang

Thank you for the submission of your manuscript to EMBO Reports. We have now received the full set of referee reports as well as referee cross-comments that are all pasted below.

As you will see, the referees acknowledge that the findings are potentially interesting. However, they do raise many concerns and queries that you would need to address.

I would thus like to invite you to revise your manuscript with the understanding that the referee concerns must be fully addressed and their suggestions taken on board. Please address all referee concerns in a complete point-by-point response. Acceptance of the manuscript will depend on a positive outcome of a second round of review. It is EMBO reports policy to allow a single round of major revision only and acceptance or rejection of the manuscript will therefore depend on the completeness of your responses included in the next, final version of the manuscript.

We realize that it is difficult to revise to a specific deadline. In the interest of protecting the conceptual advance provided by the work, we recommend a revision within 3 months (20th Aug 2025). Please discuss the revision progress ahead of this time with the editor if you require more time to complete the revisions.

- 1) A data availability section providing access to data deposited in public databases is missing. If you have not deposited any data, please add a sentence to the data availability section that explains that.
- 2) Your manuscript contains statistics and error bars based on $n=2$. Please use scatter blots in these cases. No statistics should be calculated if $n=2$.

<<https://www.embopress.org/page/journal/14693178/authorguide#expandedview>>

5) a complete author checklist, which you can download from our author guidelines

<<https://www.embopress.org/page/journal/14693178/authorguide>>. Please insert information in the checklist that is also reflected in the manuscript. The completed author checklist will also be part of the RPF.

6) Please note that all corresponding authors are required to supply an ORCID ID for their name upon submission of a revised manuscript (<<https://orcid.org/>>). Please find instructions on how to link your ORCID ID to your account in our manuscript tracking system in our Author guidelines <<https://www.embopress.org/page/journal/14693178/authorguide#authorshipguidelines>>

7) Before submitting your revision, primary datasets produced in this study need to be deposited in an appropriate public database (see <https://www.embopress.org/page/journal/14693178/authorguide#datadeposition>). Please remember to provide a reviewer password if the datasets are not yet public. The accession numbers and database should be listed in a formal "Data Availability" section placed after Materials & Method (see also <https://www.embopress.org/page/journal/14693178/authorguide#datadeposition>). Please note that the Data Availability Section is restricted to new primary data that are part of this study. * Note - All links should resolve to a page where the data can be accessed. *
If your study has not produced novel datasets, please mention this fact in the Data Availability Section.

12) All Materials and Methods need to be described in the main text using our 'Structured Methods' format, which is required for all research articles. According to this format, the Methods section includes a Reagents and Tools Table (listing key reagents, experimental models, software and relevant equipment and including their sources and relevant identifiers) followed by a Methods and Protocols section describing the methods using a step-by-step protocol format. The aim is to facilitate adoption of the methodologies across labs. More information on how to adhere to this format as well as a downloadable template (.docx) for the Reagents and Tools Table can be found in our author guidelines: <https://www.embopress.org/page/journal/14693178/authorguide#structuredmethods>.

An example of a Method paper with Structured Methods can be found here: <https://www.embopress.org/doi/full/10.1038/s44320-024-00037-6#sec-4>

I look forward to seeing a revised form of your manuscript when it is ready.

Yours sincerely,

Yehu Moran
Editor
EMBO Reports

Referee #1:

This manuscript addresses a fascinating question - the occurrence of natural cryptorchidism in cetaceans. The study presents compelling evidence of evolutionary relaxation in the *INSL3* and *RXFP2* genes among cryptorchid mammals, with cetaceans demonstrating the highest evolutionary rates. To further investigate, the authors generated *INSL3* knock-in mice and observed groin-located testes, which closely mimic the cryptorchid phenotypes seen in cetaceans and other mammals with incomplete testicular descent. Overall, the question posed and the work conducted are highly interesting and commendable.

However, one key critique lies in the rationale behind the creation of the cetacean *INSL3* knock-in mice. The authors highlight relaxed selection on cetacean *INSL3*, but they did not find an increased expression levels of this gene in cetaceans. Therefore, the justification for generating *INSL3* knock-in mice warrants clarification. Additionally, when comparing these cetacean *INSL3* knock-in mice to wild-type and *INSL3* knock-out mice, the results may not fully separate the effects of mouse *INSL3* in the wild-type group versus the sequential evolutionary changes between cetacean *INSL3* and mouse *INSL3*.

A stronger experimental approach might have been to generate *INSL3*-edit mice, replacing the mouse *INSL3* gene with cetacean-specific *INSL3*. This approach would allow for direct investigation of phenotypic effects attributable to cetacean-specific evolutionary changes. Alternatively, if a knock-in strategy is chosen, the authors should clearly articulate the reasoning behind their experimental design and how it addresses the evolutionary findings presented in the study. With these points addressed, the study would represent a robust and elegant exploration of this intriguing biological phenomenon.

Referee #2:

The authors present a molecular analysis centered on the lack of descent of the testes in mammals and specifically in cetaceans. I am not an expert on the experimental methods utilizing mice and the genes that are the focus of the study (*INSL3* and *RXFP2*), so my comments center on the authors' presentation, molecular evolutionary analyses, and inferences. An important point is that Sharma et al. (2018) cited just once in the introduction (!) clearly documented inactivation of these same two genes (*INSL3* and *RXFP2*) in multiple afrotherian mammals with testes that are not descended, and I believe they suggested multiple independent KOs of these genes within Afrotheria, so it would be good to better acknowledge the contribution of this earlier work in the introduction and in the discussion. The authors might consider the following in revising/improving their manuscript.

- 1) Line 54 and following lines. The authors note various pathways involved in descent of the testes. However, surely very few mammalian species have been examined? So, it should probably be noted that much of the knowledge of these pathways was elucidated by studies of mostly mice or humans or a few other 'model' mammalian species? Near the end of the paragraph, this is noted in part, but not earlier in the paragraph where many studies are cited and the generality of these pathways in all mammals is perhaps implied (or if not implied, this should be noted). As I am not familiar with the specific literature on testes descent and molecular manipulations relevant to studying this process, the authors should be sure that they are correct completely in their statements about causation and critical functions of different molecules (or perhaps other reviewers with more knowledge will weigh in on accuracy of this background information).
- 2) First paragraph and throughout the remaining text, English grammar and word choice/usage would benefit from much additional editing/improvement.
- 3) Line 77. It is not so easy to interpret shifts in selection to higher dN/dS values as relaxation of selection or perhaps positive selection on some sites but not others, but I trust that the authors will discriminate these alternatives later in the text. Further, since *INSL3* knockout also makes testes not descend in mice, I trust that the authors have done their experiments in such a way

that it is clear that the cetacean sequence changes in the knock-in cause the effect and not accidental KO of the gene. As noted above, my expertise is not on such experiments, so I trust the editors and other reviewers can properly assess this critical component of the paper.

4) Line 87. In the sample of 45 mammalian species, I was surprised (actually shocked?!) that no hippos were included in the molecular evolutionary analyses of dN/dS in taxa with internal, inguinal, and scrotal testes. I believe both extant hippos do not have a scrotum and their testes are 'inguinal'. Given that the focus of the paper is on the evolution of these two genes in cetaceans (with internal testes), why was their extant sister group (hippos) not included in these analyses? It is sort of odd; actually to me the most important species to sample, aside from the cetaceans? Can Hippopotamus and Choeropsis sequences be included in these analyses? If not, why or what is the justification for exclusion?

5) Figure 1. The very high dN/dS for INSL3 in Cetacea (1.02; almost exactly the 'neutral' expectation) makes me wonder whether this gene is KOed in Cetacea and whether moving this gene to mouse might therefore entail moving an inactive INSL3 gene to mouse, which has been shown to impact descent of the testes (see point #3 above). In Figure 1B, the authors note that 'evolutionary rates' are shown for two genes, but instead dN/dS which is a ratio of rates is shown. This makes me worry that the authors are not interpreting dN/dS rates correctly perhaps? In particular, in this same figure legend, the authors note lineages where there are 'accelerated rates', but this is not correct (as I understand it) because there can be deceleration of evolutionary rate in nonsynonymous and synonymous substitution, yet dN/dS might still increase. This is due to dN/dS being a ratio of two rates of evolution and not a rate of evolution. This is sort of an important point that should be addressed throughout the text. Cetacean DNA generally evolves very slowly relative to other mammals for most genes, so it is possible that the INSL3 gene is inactivated, perhaps by mutations upstream of the protein coding sequences, and there has been a lag in the accumulation of obvious frameshifts or stops in the protein-coding sequence? The authors should indicate what 'Bal' indicates in the figure, even if this is explained in the main text, the figure legend should state what this abbreviation means. In this figure, it would be interesting, perhaps, to show results in panels C, D, and E for a mouse with KOed genes for direct comparison to Bal and Mus, but again, I am not an authority on these experiments that were done with mice, so perhaps some of my comments above are not valid.

6) Line 90. Where mention the afrotherians here, the Sharma et al. (2018) results for these two genes should be noted here, since they found the genes to be KOed in multiple afrotherians, even if the afrotherians sampled here might not be KOed?

7) Line 126. It is my understanding that cetacean testes are undescended and internal, not 'inguinal' as the authors state here. Do the authors have solid cetaceans backing up the claim that cetacean testes are instead inguinal (in the inguinal canal, as in hippos)? If so, citations should be given here or earlier in the text, but I do not think this is anatomically accurate, which is a problem for the interpretation of the different experimental treatments noted in this paragraph and the following one regarding double gene treatment with the cetacean gene sequences.

8) Line 144. I am not sure how best to interpret these transcriptome results for up and down regulated genes.

9) Line 220. Again here should be careful discussing rates vs. ratio of rates (as in dN/dS), and the cetacean protein sequences look very highly conserved relative to the ancestral mammalian condition (just eyeballing the alignment) and there does not seem to be any acceleration in rate of evolution at the amino acid level within Cetacea? The mouse lineage, by contrast, has evolved a lot, so perhaps putting a very different cetacean gene into play might be expected to have a variety of diverse effects at many levels, not necessarily due to the cetacean sequence being particularly fast evolving but because an 'alien' cetacean gene sequence that differs very much from the fast evolving mouse amino acid sequence has been placed on a very distant mouse genetic background. So, I think the authors should be very circumspect in their interpretations of what is going on?

Referee #3:

This manuscript describes how natural cryptorchidism is induced at the gene level. The authors show that both INSL3 and RXFP2 exhibited the highest evolutionary rates in cetaceans, a naturally cryptorchid mammals, compared to those in scrotal mammals. When primarily cultured mouse gubernacular cells were treated with cetacean INSL, cAMP-PKA-CREB pathway was down regulated. By generating cetacean INSL3 gene knock-in mice, the authors show that the knock-in mice exhibited cryptorchid phenotypes with incompletely descended testes. Moreover, these mice displayed male sterility, impaired testicular development, and upregulated inflammatory pathways in testes by RNA-seq analysis.

The work represents significant contribution to the better understanding of cryptorchidism in mammals, and influences the field of reproductive biology and evolutionary biology. The experiments are well conducted. However, the manuscript writing can be improved to convince readers that the work is novel, and that authors' logic, flow, and their conclusions are correct. Below I list some comments about presentation of data or manuscript writing.

1) The authors show that Cetacean INSL3 knock-in mice mimicked cryptorchid phenotypes in cetaceans and the other mammals with incompletely descended testes. However, it is unclear if the Cetacean INSL3 is still functional or become non-functional during the course of evolution. The authors should discuss this point, citing amino acid sequence alignment data (Fig.S4) and/or performing additional experiments.

2) Likewise, it is unclear if Cetacean INSL3 knock-in mice just mimicked the phenotypes found in INSL3 KO mice. To address this question, the authors should compare the transcriptome between INSL3 knock-in mice and INSL3 KO mice in figs 4 and 6.

3) Results (lines 205-209) and Discussion (lines 285-289): Because the RNA-seq analysis exhibited the upregulation of inflammatory pathways (Fig.6D), the authors have suggested that the inflammation in the cryptorchid testes of INSL3-KI mice may impair spermatogenesis.

However, the authors do not include any evidence or references to ensure that the testicular inflammation induces severe defect of spermatogenesis in cryptorchid testes. The authors should cite the references in lines 288-289. Moreover, the undescended testes experience a higher temperature than a normally descended testes; the high temperature can induce meiotic failures during spermatogenesis (Hirano et al, 2022, DOI: 10.1038/s42003-022-03449-y). The authors should show gene expression patterns involved in heat stress and meiosis by using the data set in Fig.6.

4) Results (lines 175-180): The data to ensure the following sentences should be added to the Figures:

"The testicular structure of the seminiferous tubules (STs) was assessed and classified as normal (all germ cell layers present and correctly positioned), abnormal (germ cell loss or misplacement), or Sertoli cell-only (SCO, no germ cells). No variation in the percentage of normal seminiferous tubules was observed in WT mice across all ages. However, by 3 weeks of age, a few abnormal seminiferous tubules were already observable in the testes of KO mice. In addition, seminiferous tubule abnormalities in INSL3-KI mice did not appear until 1.5 months, at which point nearly all tubules in KO mice were already abnormal"

5) Fig.5C: The labels "immature" and "sexually mature" are ambiguous. The age of mice should be added in the Figure panel.

6) Results (lines 189-191): The authors state that analysis using the computer-assisted sperm analysis (CASA) system further revealed a severely reduced sperm motility in INSL3-KI mice (Fig. 5D). However, epididymal histology image in Fig. 5C does not show any sperm. The authors should specify the source of sperm used for CASA system.

7) Figure legend (for Fig. 1C, 1D, 1E, 2A, and 2B): The authors should clarify what the terms "Mus" and "Bal" stand for.

Dear Dr. Yehu Moran,

We thank you and the reviewers for the constructive feedback on our manuscript entitled "*Evolutionary Relaxation and Functional Changes of INSL3/RXFP2 in Natural Cryptorchidism of Mammals*" (ID: EMBOR-2025-61492V2).

We are grateful for the opportunity to revise our work for EMBO Reports. In this revision, we have carefully addressed all referee concerns through additional analyses, new data, revised figures, and clarified text according to the journal's requirements. Raw RNA-seq data have been deposited in the NCBI Gene Expression Omnibus (GEO) under accession number **GSE305702**. Reviewer access has been provided via a private token, and the dataset will be made publicly accessible upon publication. The Materials and Methods have been reformatted into the Structured Methods style, and Expanded View/Appendix figures have been prepared as requested. We believe these revisions have substantially strengthened the manuscript and fully addressed the reviewers' and editorial requirements. Given the breadth of data and the number of main figures, we wish to resubmit our work as a Full Article.

Revisions include:

1. Additional analyses and new data.
2. Revised text for clarity and conciseness.
3. Updated figures and supplementary information where appropriate.

Below, we reproduce each comment in full, followed by our detailed responses. All changes are highlighted in the revised manuscript.

Referee #1:

This manuscript addresses a fascinating question - the occurrence of natural cryptorchidism in cetaceans. The study presents compelling evidence of evolutionary relaxation in the INSL3 and RXFP2 genes among cryptorchid mammals, with cetaceans demonstrating the highest evolutionary rates. To further investigate, the authors generated INSL3 knock-in mice and observed groin-located testes, which closely mimic the cryptorchid phenotypes seen in cetaceans and other mammals with incomplete testicular descent. Overall, the question posed and the work conducted are highly interesting and commendable.

However, one key critique lies in the rationale behind the creation of the cetacean INSL3 knock-in mice. The authors highlight relaxed selection on cetacean INSL3, but they did not find an increased expression levels of this gene in cetaceans. Therefore, the justification for generating INSL3 knock-in mice warrants clarification. Additionally, when comparing these cetacean INSL3 knock-in mice to wild-type and INSL3 knock-out mice, the results may not fully separate the effects of mouse INSL3 in the wild-type group versus the sequential evolutionary changes between cetacean INSL3 and mouse INSL3.

A stronger experimental approach might have been to generate INSL3-edit mice, replacing the mouse INSL3 gene with cetacean-specific INSL3. This approach would allow for direct investigation of phenotypic effects attributable to cetacean-specific evolutionary changes. Alternatively, if a knock-in strategy is chosen, the authors should clearly articulate the reasoning

behind their experimental design and how it addresses the evolutionary findings presented in the study. With these points addressed, the study would represent a robust and elegant exploration of this intriguing biological phenomenon.

Response:

We thank the reviewer for the overall positive evaluation and for raising these important points regarding our experimental design.

The KI model is not an overexpression line; the mouse *Insl3* coding sequence was replaced with the cetacean sequence at the native locus, preserving all regulatory elements to ensure physiological expression. This design matches the “replacement” approach suggested by the reviewer and allows direct testing of cetacean ligand function *in vivo*.

Although expression levels were not elevated, evolutionary analyses indicated strong relaxed selection, suggesting functional divergence at the peptide level. The KI model was therefore used to assess whether these sequence changes contribute to cryptorchid phenotypes.

We have clarified this rationale in the Results (lines 129–133) and noted its limitations in the Discussion (lines 300–308).

Referee #2:

The authors present a molecular analysis centered on the lack of descent of the testes in mammals and specifically in cetaceans. I am not an expert on the experimental methods utilizing mice and the genes that are the focus of the study (*INSL3* and *RXFP2*), so my comments center on the authors' presentation, molecular evolutionary analyses, and inferences. An important point is that Sharma et al. (2018) cited just once in the introduction (!) clearly documented inactivation of these same two genes (*INSL3* and *RXFP2*) in multiple afrotherian mammals with testes that are not descended, and I believe they suggested multiple independent KOs of these genes within Afrotheria, so it would be good to better acknowledge the contribution of this earlier work in the introduction and in the discussion. The authors might consider the following in revising/improving their manuscript.

Response:

We thank the reviewer for the thoughtful and constructive comments. We appreciate the recognition of our model's novelty and the suggestions that helped improve the clarity and scientific rigor of the manuscript.

We have now cited Sharma et al. (2018) in the Introduction (lines 74–77) to highlight the link between *INSL3/RXFP2* disruption and undescended testes across Afrotherian lineages, strengthening the evolutionary context of our study.

1) Line 54 and following lines. The authors note various pathways involved in descent of the testes. However, surely very few mammalian species have been examined? So, it should probably be noted that much of the knowledge of these pathways was elucidated by studies of mostly mice

or humans or a few other 'model' mammalian species? Near the end of the paragraph, this is noted in part, but not earlier in the paragraph where many studies are cited and the generality of these pathways in all mammals is perhaps implied (or if not implied, this should be noted). As I am not familiar with the specific literature on testes descent and molecular manipulations relevant to studying this process, the authors should be sure that they are correct completely in their statements about causation and critical functions of different molecules (or perhaps other reviewers with more knowledge will weigh in on accuracy of this background information).

Response:

In the Introduction (lines 52–55), we have revised the text to state that mechanistic insights into testicular descent are largely derived from model species, and that the assumption of pathway conservation across mammals should be interpreted with caution.

2) First paragraph and throughout the remaining text, English grammar and word choice/usage would benefit from much additional editing/improvement.

Response:

We have revised the entire manuscript for grammar, clarity, and style; the text was polished by a professional language editing service.

3) Line 77. It is not so easy to interpret shifts in selection to higher dN/dS values as relaxation of selection or perhaps positive selection on some sites but not others, but I trust that the authors will discriminate these alternatives later in the text. Further, since *INSL3* knockout also makes testes not descend in mice, I trust that the authors have done their experiments in such a way that it is clear that the cetacean sequence changes in the knock-in cause the effect and not accidental KO of the gene. As noted above, my expertise is not on such experiments, so I trust the editors and other reviewers can properly assess this critical component of the paper.

Response:

Regarding the interpretation of elevated dN/dS ratios, we agree that such shifts can reflect either relaxation of purifying selection or episodic positive selection acting on a subset of sites. To address this, we conducted additional analyses using the RELAX framework (Wertheim et al. 2015), which explicitly tests for relaxation versus intensification of selection. These results support a significant relaxation of selective pressure on *INSL3* in cryptorchid lineages, including cetaceans.

With respect to the knock-in model, we fully agree that it is essential to ensure that the phenotype observed in *INSL3*-KI mice is specifically due to the expression of cetacean *INSL3*, rather than a loss-of-function artifact. To this end, we carefully validated the knock-in allele to confirm appropriate expression levels of the cetacean transgene in the testis, and verified the absence of premature stop codons or frameshifts that would indicate a non-functional allele. These results support the conclusion that the observed phenotype arises from the sequence features of cetacean *INSL3*, not from a gene disruption. This point is already addressed in the Methods (Lines 402). We have now uploaded the original sequencing data confirming that the KI allele contains no

premature stop codons.

4) Line 87. In the sample of 45 mammalian species, I was surprised (actually shocked?!) that no hippos were included in the molecular evolutionary analyses of dN/dS in taxa with internal, inguinal, and scrotal testes. I believe both extant hippos do not have a scrotum and their testes are 'inguinal'. Given that the focus of the paper is on the evolution of these two genes in cetaceans (with internal testes), why was their extant sister group (hippos) not included in these analyses? It is sort of odd; actually to me the most important species to sample, aside from the cetaceans? Can Hippopotamus and Choeropsis sequences be included in these analyses? If not, why or what is the justification for exclusion?

Response:

We added *Hippopotamus amphibius* INSL3 and RXFP2 to update dN/dS and RELAX analyses, showing elevated ratios and relaxed selection similar to cetaceans (Results, lines 94–101; Methods, line 410). *Choeropsis liberiensis* has identical coding sequences and was excluded to avoid redundancy. Figure 1 and Table EV1 have been updated.

5) Figure 1. The very high dN/dS for INSL3 in Cetacea (1.02; almost exactly the 'neutral' expectation) makes me wonder whether this gene is KOed in Cetacea and whether moving this gene to mouse might therefore entail moving an inactive INSL3 gene to mouse, which has been shown to impact descent of the testes (see point #3 above). In Figure 1B, the authors note that 'evolutionary rates' are shown for two genes, but instead dN/dS which is a ratio of rates is shown. This makes me worry that the authors are not interpreting dN/dS rates correctly perhaps? In particular, in this same figure legend, the authors note lineages where there are 'accelerated rates', but this is not correct (as I understand it) because there can be deceleration of evolutionary rate in nonsynonymous and synonymous substitution, yet dN/dS might still increase. This is due to dN/dS being a ratio of two rates of evolution and not a rate of evolution. This is sort of an important point that should be addressed throughout the text. Cetacean DNA generally evolves very slowly relative to other mammals for most genes, so it is possible that the INSL3 gene is inactivated, perhaps by mutations upstream of the protein coding sequences, and there has been a lag in the accumulation of obvious frameshifts or stops in the protein-coding sequence? The authors should indicate what 'Bal' indicates in the figure, even if this is explained in the main text, the figure legend should state what this abbreviation means. In this figure, it would be interesting, perhaps, to show results in panels C, D, and E for a mouse with KOed genes for direct comparison to Bal and Mus, but again, I am not an authority on these experiments that were done with mice, so perhaps some of my comments above are not valid.

Response:

For INSL3 in cetaceans (dN/dS = 1.02), coding-sequence analysis revealed no frameshifts, premature stops, or loss of conserved motifs, arguing against pseudogenization. In vitro assays and KI mouse phenotypes differ from knockouts, indicating retained functionality. We now replace "evolutionary rate" with "dN/dS ratio" throughout, describe branches as showing "relaxed selection," and clarify in Figure 1 legend that dN/dS reflects selective pressure (lines 786). "Bal" and "Mus" in Figure 1 are now defined (lines 796–799). Panels C–E represent cell-based assays,

where the equivalent of the KO mouse condition is the absence of exogenous INSL3 protein (blank control). This control was included in the data analysis to define the baseline response, but is not shown in the final figure for clarity.

6) Line 90. Where mention the afrotherians here, the Sharma et al. (2018) results for these two genes should be noted here, since they found the genes to be KOed in multiple afrotherians, even if the afrotherians sampled here might not be KOed?

Response:

In the revised manuscript, we now clearly note that although Sharma et al. (2018) reported pseudogenization of INSL3 and RXFP2 in several testicond Afrotherians (e.g., *Trichechus manatus*, *Loxodonta africana*), they also indicated that not all Afrotherians exhibit such gene loss. For our cross-mammalian evolutionary analyses, we specifically selected Afrotherian species with intact coding sequences for both genes to enable reliable dN/dS estimation. Notably, these intact-gene Afrotherians also exhibited relaxed selection patterns similar to those of cetaceans, reinforcing our conclusion that reduced selective constraints on INSL3 and RXFP2 are associated with abdominal testes even in the absence of pseudogenization.

7) Line 126. It is my understanding that cetacean testes are undescended and internal, not 'inguinal' as the authors state here. Do the authors have solid cetaceans backing up the claim that cetacean testes are instead inguinal (in the inguinal canal, as in hippos)? If so, citations should be given here or earlier in the text, but I do not think this is anatomically accurate, which is a problem for the interpretation of the different experimental treatments noted in this paragraph and the following one regarding double gene treatment with the cetacean gene sequences.

Response:

We revised the text (lines 138–142) to state that cetacean testes are intra-abdominal, located dorsally and caudally, posterior to the kidneys (Rommel et al., 1992), correcting the earlier implication of an inguinal position while preserving the intended phenotypic comparison.

8) Line 144. I am not sure how best to interpret these transcriptome results for up and down regulated genes.

Response:

Our intention was to emphasize that the transcriptomic changes in INSL3-KI mice form a coherent biological sequence rather than a set of unrelated gene lists. Specifically, in the gubernaculum at P8, most downregulated genes were enriched in pathways related to muscle development and contraction, indicating impaired structural remodeling required for testicular descent. In the testes, we observed that upregulated genes were predominantly associated with inflammatory and immune activation pathways, whereas downregulated genes included key regulators of spermatogenesis, meiosis, and cell adhesion. These transcriptional patterns are consistent with the observed phenotypes: early gubernacular dysfunction, followed by progressive testicular degeneration and reduced fertility in KI males. We have revised the Results section to integrate these findings into a single explanatory framework and improve the interpretability of the up- and

downregulated gene categories.

9) Line 220. Again here should be careful discussing rates vs. ratio of rates (as in dN/dS), and the cetacean protein sequences look very highly conserved relative to the ancestral mammalian condition (just eyeballing the alignment) and there does not seem to be any acceleration in rate of evolution at the amino acid level within Cetacea? The mouse lineage, by contrast, has evolved a lot, so perhaps putting a very different cetacean gene into play might be expected to have a variety of diverse effects at many levels, not necessarily due to the cetacean sequence being particularly fast evolving but because an 'alien' cetacean gene sequence that differs very much from the fast evolving mouse amino acid sequence has been placed on a very distant mouse genetic background. So, I think the authors should be very circumspect in their interpretations of what is going on?

Response:

While the alignment in Figure EV3 visually suggests that cetacean INSL3 is relatively conserved, this impression is partly influenced by the high proportion of cetacean sequences in the figure and the limited representation of rapidly evolving non-cetacean lineages, such as rodents. As a result, mouse sequences may appear disproportionately divergent. Our codon-based PAML analyses, which incorporate a balanced taxonomic sampling across mammals, indicate that cetacean INSL3 shows elevated dN/dS values consistent with relaxed selection, rather than extensive amino acid divergence. We agree that interspecies knock-in approaches inherently carry the risk of functional incompatibilities unrelated to specific amino acid substitutions in the ligand, and we have now explicitly acknowledged both of these caveats in the revised Discussion.

Referee #3:

This manuscript describes how natural cryptorchidism is induced at the gene level. The authors show that both INSL3 and RXFP2 exhibited the highest evolutionary rates in cetaceans, a naturally cryptorchid mammals, compared to those in scrotal mammals. When primarily cultured mouse gubernacular cells were treated with cetacean INSL, cAMP-PKA-CREB pathway was down regulated. By generating cetacean INSL3 gene knock-in mice, the authors show that the knock-in mice exhibited cryptorchid phenotypes with incompletely descended testes. Moreover, these mice displayed male sterility, impaired testicular development, and upregulated inflammatory pathways in testes by RNA-seq analysis.

The work represents significant contribution to the better understanding of cryptorchidism in mammals, and influences the field of reproductive biology and evolutionary biology. The experiments are well conducted. However, the manuscript writing can be improved to convince readers that the work is novel, and that authors' logic, flow, and their conclusions are correct. Below I list some comments about presentation of data or manuscript writing.

Response:

We thank the reviewer for the positive assessment of our study and for recognizing its contribution to understanding the genetic basis of natural cryptorchidism in mammals. We appreciate the thoughtful and constructive comments, which have helped us improve the clarity, interpretation, and presentation of our results. Below, we provide a detailed point-by-point response.

1) The authors show that Cetacean INSL3 knock-in mice mimicked cryptorchid phenotypes in cetaceans and the other mammals with incompletely descended testes. However, it is unclear if the Cetacean INSL3 is still functional or become non-functional during the course of evolution. The authors should discuss this point, citing amino acid sequence alignment data (Fig.S4) and/or performing additional experiments.

Response:

As shown in Figure EV3, cetacean INSL3 retains all four structural domains, with receptor-binding A- and B-chains highly conserved. Most substitutions occur in cleaved regions (signal peptide, C-chain). Combined with our in vitro data showing reduced but detectable cAMP–PKA–CREB activation, this supports partial functionality rather than pseudogenization (Lines 284–288).

2) Likewise, it is unclear if Cetacean INSL3 knock-in mice just mimicked the phenotypes found in INSL3 KO mice. To address this question, the authors should compare the transcriptome between INSL3 knock-in mice and INSL3 KO mice in figs 4 and 6.

Response:

We performed RNA-seq on INSL3-KO mice (P8 gubernaculum; P8, P23 testis) and compared with KI and WT. KO vs. WT yielded more DEGs than KI vs. WT. KO- and KI-shared downregulation in P8 gubernaculum involved muscle/cytoskeleton, but KO-specific Notch/AMPK downregulation suggests partial KI rescue. In P23 testis, KO uniquely showed stronger spermatogenesis downregulation and inflammatory activation absent in KI. These results indicate partial cetacean INSL3 activity. Figures 4 and 6, and the corresponding Results section, have been reorganized accordingly.

3) Results (lines 205-209) and Discussion (lines 285-289): Because the RNA-seq analysis exhibited the upregulation of inflammatory pathways (Fig.6D), the authors have suggested that the inflammation in the cryptorchid testes of INSL3-KI mice may impair spermatogenesis.

However, the authors do not include any evidence or references to ensure that the testicular inflammation induces severe defect of spermatogenesis in cryptorchid testes. The authors should cite the references in lines 288-289. Moreover, the undescended testes experience a higher temperature than a normally descended testes; the high temperature can induce meiotic failures during spermatogenesis (Hirano et al, 2022, DOI: 10.1038/s42003-022-03449-y). The authors should show gene expression patterns involved in heat stress and meiosis by using the data set in Fig.6.

Response:

We added references (Aldahhan et al., 2021; Hirano et al., 2022; Wang et al., 2024). We analyzed heat shock (e.g., *Hspa1b*, *Hsp90ab1*) and meiosis genes (e.g., *Rec8*, *Cpeb3*) from RNA-seq. We added Fig. 6F and discussed the potential evolved inflammation control in natural cryptorchid species.

4) Results (lines 175-180): The data to ensure the following sentences should be added to the

Figures:

"The testicular structure of the seminiferous tubules (STs) was assessed and classified as normal (all germ cell layers present and correctly positioned), abnormal (germ cell loss or misplacement), or Sertoli cell-only (SCO, no germ cells). No variation in the percentage of normal seminiferous tubules was observed in WT mice across all ages. However, by 3 weeks of age, a few abnormal seminiferous tubules were already observable in the testes of KO mice. In addition, seminiferous tubule abnormalities in INSL3-KI mice did not appear until 1.5 months, at which point nearly all tubules in KO mice were already abnormal"

Response:

We added a new panel (Fig. 5D) quantifying normal, abnormal, and SCO tubules, which confirms earlier degeneration in KO compared with delayed onset in KI. We updated the legend (Lines 832) and the Results section (Lines 214–217) accordingly.

5) Fig.5C: The labels "immature" and "sexually mature" are ambiguous. The age of mice should be added in the Figure panel.

Response:

We replaced "immature" and "sexually mature" with "P20" and "P60" in Fig. 5C and its legend.

6) Results (lines 189-191): The authors state that analysis using the computer-assisted sperm analysis (CASA) system further revealed a severely reduced sperm motility in INSL3-KI mice (Fig. 5D). However, epididymal histology image in Fig. 5C does not show any sperm. The authors should specify the source of sperm used for CASA system.

Response:

As stated in the Methods and in the legend of Fig. 5E (Lines 845), sperm for CASA were collected from the cauda epididymis of 2-month-old mice. The histology in Fig. 5C shows a limited section and does not capture the entire cauda. We clarified that reduced sperm output in INSL3-KI mice also explains the absence of visible sperm in the displayed section despite their detection for CASA.

7) Figure legend (for Fig. 1C, 1D, 1E, 2A, and 2B): The authors should clarify what the terms "Mus" and "Bal" stand for.

Response:

We clarified in the figure legends that "Bal" refers to *Balaenoptera acutorostrata* INSL3 and "Mus" refers to mouse INSL3.

Please let us know if further clarification is required. We believe that the additional experiments, new analyses, and clarifications fully address the reviewers' concerns. We hope the revised manuscript now meets the standards for publication in *EMBO Reports*.

Sincerely,

Guang Yang

Dear Prof. Yang

Thank you for the submission of your revised manuscript to our offices. We have now received the enclosed reports from the original referees that were asked to assess it. EMBOR-2025-61492V2 still has minor suggestions that I would like you to incorporate before we can proceed with the official acceptance of your manuscript.

Additionally, our editorial assistance team provided more technical and formatting comments that must be fully addressed to allow final acceptance of the manuscript.

I look forward to seeing a new revised version of your manuscript as soon as possible.

Yehu Moran
Academic Editor
EMBO Reports

comments by editorial assistance team

MANUSCRIPT FORMAT: Error - It has main text and figures; the figures need to be removed and we only need main figure legends and EV figure legends provided at the end of the text with the figures being provided only via the system as separate files.

Conflict of Interest: included, but it needs to be renamed to Disclosure and Competing Interests Statement

AUTHORS: name discrepancy - Shixia Xu in the ms vs. Xia Shi Xu in the system; we need the corresponding author's email address on the title page.

Author Contribution/CRedit: need to be removed from the text and provided only via the submission system.

FUNDING INFO: missing in the system - MOST grant number U24A20362, PI Project of Southern Marine Science and Engineering Guangdong Laboratory (Guangzhou) (GML2021 GD0805), the Qing Lan Project of Jiangsu Province, and the Priority Academic Program Development of Jiangsu Higher Education Institutions; Funding should be part of Acknowledgments section so no separate section heading is needed in the text.

FIGURES IN SEPARATE FILES: yes for main figures, but EV figures are only provided in the ms - we also need individual and production quality EV figures.

FIGURE CALLOUTS: missing a callout for Figure 5E in text. Please correct.

DATASET EV LEGENDS: Table EV1 provided in the manuscript; EV tables should be provided as separate files (file type Expanded View Content); if a table needs to remain in the manuscript text, then it should be called Table 1 and be placed between main and EV figure legends.

APPENDIX FILE WITH ToC: in, but page numbers are needed throughout the file and on the title page to show where each item is located. Please correct.

SYNOPSIS IMAGE: missing. Please provide.

SYNOPSIS TEXT: missing. Please provide.

EXTRA NOTES:

- Materials and Methods should be named Methods.
- The manuscript sections should be in the following order: Title page - Abstract & Keywords - Introduction - Results - Discussion - Methods - Data Availability - Acknowledgments - Disclosure Statement & Competing Interests - References - Figure Legends - (Main Tables with legends if applicable) - Expanded View Figure Legends.

*Please note that the specific URL for GSE305702 dataset needs to be provided in the data availability statement.

Figure Legends - Comments

- Please define the annotated p values ****/**/* as well as provide the exact p-values for the same in the legend of figure EV2 as appropriate.
- Please note that the exact p values are not provided in the legends of figures 1C-E; 2A, B; 3B, 5A, E. Please provide.
- Please indicate the statistical test used for data analysis in the legend of figure EV2
- Please note that information related to n is missing in the legend of figure EV2. Please provide.
- Please note that the error bars are not defined in the legends of figures 3A, 5A, B, E; EV2. Please define.

comments by referees

Referee #1:

Thank you for the thorough revisions; most of my earlier concerns have been addressed. I have a few minor, primarily clarificatory suggestions to further strengthen the manuscript:

Since the INSL3/RXFP2-DKI mouse model may not fully recapitulate phenotypes observed across naturally cryptorchid

(testicond) mammals. In addition, the molecular mechanism by which the identified substitutions contribute to cryptorchidism has not been directly examined. To reflect these limitations, I recommend softening the overarching claims and, if acceptable, adopting a more cautious title, such as: "Evolutionary Relaxation and Functional Shifts in the INSL3-RXFP2 Axis May Underlie Natural Cryptorchidism in Mammals."

Please verify the ω values on the foreground branch. As currently shown, the foreground ω appears lower than the background ω , which is at odds with the text stating it is higher.

Line 158 (P8 and P40): Please clarify what "P8" and "P40" refer to.

Discussion (A- and B-chain conservation): The text states that changes in cetaceans are largely confined to the signal peptide and C-chain while A/B chains are highly conserved. However, Figure EV3 appears to show a Valine in the A chain that is conserved across cetaceans and other naturally cryptorchid mammals. Please verify this residue and reconcile with the statement on A/B-chain conservation. If it is a genuine, potentially convergent substitution, consider discussing its possible implications.

Referee #2:

I believe that the authors have adequately responded to my various comments in my first round of review.

Referee #3:

The authors have responded thoroughly and carefully to the most of my concerns, and the manuscript has greatly improved. There is only a remaining issue, as follow:

To improve reliability of CASA data and the authors' work (Fig. 5E), the authors should add a new graph figure that shows the proportion of epididymal tubules (%) containing sperm or no sperm at P60, in WT, INSL3-KI, and INSL3-KO mice.

Dear Dr. Yehu Moran,

We thank you and the reviewers for the constructive feedback on our manuscript entitled “*Evolutionary Relaxation and Functional Change of INSL3 and RXFP2 May Underlie Natural Cryptorchidism in Mammals*” (ID: EMBOR-2025-61492V3).

We are grateful for the opportunity to revise our work for EMBO Reports. In this revision, we have carefully addressed the remaining referee comments and fully implemented the technical and formatting requirements from the editorial assistance team. The changes primarily involve responding to the reviewers' remaining points, refining the text, correcting figure and legend inconsistencies, and making formatting adjustments to comply with production requirements, including the addition of the required synopsis image and text.

Below, we reproduce each comment in full, followed by our detailed responses. All changes are highlighted in the revised manuscript.

Editorial Assistance Comments

We thank the editorial team for the detailed technical guidance. All issues raised have been fully addressed as follows:

1. **Figure and manuscript structure:** All figures were removed from the main text and uploaded separately. Main and EV figure legends are now placed at the end of the manuscript.
2. **Section titles:** “Conflict of Interest” was renamed to “Disclosure and Competing Interests Statement.” “Materials and Methods” was renamed to “Methods.”
3. **Author information:** The corresponding author's name in the submission system was corrected from “Xia Shi Xu” to “Shixia Xu,” and the corresponding author's email address was added to the title page.
4. **Author contributions:** The author contribution statement was removed from the text and entered only in the submission system.
5. **Funding information:** Funding details were moved to the Acknowledgments section and formatted according to journal requirements.
6. **EV files:** All Expanded View figures and tables were uploaded as individual production-quality files, and the Table EV1 originally included in the manuscript has also been provided as a separate file.
7. **Text and callouts:** A missing callout for Figure 5E was added, and figure numbering was verified throughout the text.
8. **Appendix file with ToC:** Page numbers have been added throughout the appendix file and on the title page.
9. **Synopsis materials:** A synopsis image and a short synopsis text were provided via the submission system.
10. **Formatting and order:** The manuscript structure was reorganized to follow EMBO guidelines.
11. **Figure legends:** All legends were revised to define p-value annotations (*, **, ***, ****),

provide exact p-values, indicate statistical tests, include n values, and define error bars.

Referee #1:

1) Since the INSL3/RXFP2-DKI mouse model may not fully recapitulate phenotypes observed across naturally cryptorchid (testicond) mammals. In addition, the molecular mechanism by which the identified substitutions contribute to cryptorchidism has not been directly examined. To reflect these limitations, I recommend softening the overarching claims and, if acceptable, adopting a more cautious title, such as: "Evolutionary Relaxation and Functional Shifts in the INSL3-RXFP2 Axis May Underlie Natural Cryptorchidism in Mammals."

Response:

We fully agree with the reviewer that the original title may overstate the causal link between molecular signatures and natural cryptorchidism. We have revised the title to a more cautious formulation that better reflects the scope of our study. The new title reads: "Evolutionary Relaxation and Functional Change of INSL3 and RXFP2 May Underlie Natural Cryptorchidism in Mammals"

2) Please verify the ω values on the foreground branch. As currently shown, the foreground ω appears lower than the background ω , which is at odds with the text stating it is higher.

Response:

We thank the reviewer for catching this error. The foreground and background ω values were inadvertently swapped in Table EV1. We have corrected this mistake in Table EV1. The interpretation of the results remains unchanged.

3) Line 158 (P8 and P40): Please clarify what "P8" and "P40" refer to.

Response:

We have clarified in the text that "P8" and "P40" refer to postnatal day 8 and postnatal day 40, respectively, and have checked all other time point abbreviations to ensure that each is defined upon its first appearance in the manuscript.

4) Discussion (A- and B-chain conservation): The text states that changes in cetaceans are largely confined to the signal peptide and C-chain while A/B chains are highly conserved. However, Figure EV3 appears to show a Valine in the A chain that is conserved across cetaceans and other naturally cryptorchid mammals. Please verify this residue and reconcile with the statement on A/B-chain conservation. If it is a genuine, potentially convergent substitution, consider discussing its possible implications.

Response:

Upon re-examining the alignment, we confirmed that the Valine residue in the A-chain is present in cetaceans and several rodent species (e.g., mouse, rat), as well as naked mole-rat. However, this substitution is not shared across all naturally cryptorchid taxa and is therefore unlikely to represent a convergent adaptation. We have revised the text to clarify that the A- and B-chains are overall

highly conserved, with a few lineage-specific substitutions (e.g., Valine in rodents and cetaceans), without functional or evolutionary claims.

Referee #2

I believe that the authors have adequately responded to my various comments in my first round of review.

Response:

We thank the reviewer for their positive assessment and are pleased that no further concerns were raised at this stage.

Referee #3

To improve reliability of CASA data and the authors' work (Fig. 5E), the authors should add a new graph figure that shows the proportion of epididymal tubules (%) containing sperm or no sperm at P60, in WT, INSL3-KI, and INSL3-KO mice.

Response:

We thank the reviewer for this constructive suggestion aimed at strengthening the reliability of our CASA data. As requested, we have quantified the proportion of epididymal tubules containing sperm versus those lacking sperm in WT, INSL3-KI, and INSL3-KO mice at postnatal day 60 (P60).

The results are now presented as a new panel Figure 5F, showing a significantly lower proportion of sperm-containing epididymal tubules in INSL3-KI and INSL3-KO mice compared to WT controls.

We hope that these revisions and clarifications fully address the remaining concerns and bring the manuscript closer to acceptance at *EMBO Reports*. We thank the editorial team and reviewers for their time and consideration, and we look forward to a positive decision.

Sincerely,

Guang Yang

Prof. Guang Yang
Nanjing Normal University
School of Life Sciences
Xingzhi Building, Xianlin Campus, Nanjing Normal University, No.1 Wenyuan Road, Qixia District, Nanjing City
Nanjing, Jiangsu Province 210046
China

Dear Prof. Yang,

I am very pleased to accept your manuscript for publication in the next available issue of EMBO Reports. Thank you for your contribution to our journal.

Just one last point to be made clear: all data must become now publicly available including GSE305702. Please make sure this is done as soon as possible and inform us via email afterwards, otherwise we would not be able to proceed with the publication of your paper.

Yours sincerely,

Yehu Moran
Academic Editor
EMBO Reports
